# RNA splicing programs define tissue compartments and cell types at single-cell resolution

**Julia Eve Olivieri[1,2,3]†, Roozbeh Dehghannasiri[2,3]†, Peter L Wang[3], SoRi Jang[3], Antoine de Morree[4], Serena Y Tan[5], Jingsi Ming[6,7], Angela Ruohao Wu[8], Tabula Sapiens Consortium, Stephen R Quake[9,10], Mark A Krasnow[3], Julia Salzman[2,3]***

[1]Institute for Computational and Mathematical Engineering, Stanford University, Stanford, United States; [2]Department of Biomedical Data Science, Stanford University, Stanford, United States; [3]Department of Biochemistry, Stanford University, Stanford, United States; [4]Department of Neurology and Neurological Sciences, Stanford University School of Medicine, Stanford, United States; [5]Department of Pathology, Stanford University Medical Center, Stanford, United States; [6]Academy for Statistics and Interdisciplinary Sciences, Faculty of Economics and Management,East China Normal University, Shanghai, China; [7]Department of Mathematics, The Hong Kong University of Science and Technology, Hong Kong, China; [8]Department of Chemical and Biological Engineering, The Hong Kong University of Science and Technology, Hong Kong, China; [9]Chan Zuckerberg Biohub, San Francisco, United States; [10]Department of Bioengineering, Stanford University, Stanford, United States

**\*For correspondence:**
julia.salzman@stanford.edu

†These authors contributed equally to this work

**Group author details:**
Tabula Sapiens Consortium See page 23

**Competing interest:** The authors declare that no competing interests exist.

**Abstract** The extent splicing is regulated at single-cell resolution has remained controversial due to both available data and methods to interpret it. We apply the SpliZ, a new statistical approach, to detect cell-type-specific splicing in >110K cells from 12 human tissues. Using 10X Chromium data for discovery, 9.1% of genes with computable SpliZ scores are cell-type-specifically spliced, including ubiquitously expressed genes *MYL6* and *RPS24*. These results are validated with RNA FISH, single-cell PCR, and Smart-seq2. SpliZ analysis reveals 170 genes with regulated splicing during human spermatogenesis, including examples conserved in mouse and mouse lemur. The SpliZ allows model-based identification of subpopulations indistinguishable based on gene expression, illustrated by subpopulation-specific splicing of classical monocytes involving an ultraconserved exon in *SAT1*. Together, this analysis of differential splicing across multiple organs establishes that splicing is regulated cell-type-specifically.

## Introduction

Isoform-specific RNA expression is conserved in higher eukaryotes (*Merkin et al., 2012*), tissue-specific, and controls developmental (*Baralle and Giudice, 2017*; *Keren et al., 2010*; *Ule and Blencowe, 2019*; *Zhang et al., 2016*) and myriad cell signaling pathways (*Hartmann et al., 2009*; *Martinez et al., 2012*). Alternative splicing also plays a major functional role as it expands proteomic complexity and rewires protein interaction networks (*Buljan et al., 2012*; *Ellis et al., 2012*). Alternative RNA isoforms of the same gene can even be translated into proteins with opposite functions (*Yang et al., 2016*). Splicing is dysregulated in many diseases from neurological disorders to cancers (*Anczuków and Krainer, 2016*). Alternative splicing studies have been mostly limited to bulk-level analysis, and they have shown evidence that as many as one-third of all human genes express tissue-dependent dominant isoforms, while most highly expressed human genes express a single dominant

isoform in different tissues (*Ezkurdia et al., 2015*; *Gonzàlez-Porta et al., 2013*). It has been known for decades that genes can have cell-type-specific splicing patterns, best characterized in the immune, muscle, and nervous systems (*Florea et al., 2013*; *Giudice et al., 2016*; *Li et al., 2007*; *Martinez et al., 2012*; *Zipursky and Sanes, 2010*). But the extent of cell-type-specific splicing is still controversial, partly because it has only been studied indirectly through profiling tissues, which is confounded by differential cell type composition. Many other questions remain such as whether cells of the same type in different tissues have shared splicing programs.

Determining how splicing is regulated in single cells could improve predictive models of splice isoform expression and move toward systems-level prediction of function. Furthermore, single-cell RNA splicing analysis has tremendous implications for biomedicine. Drugs targeting 'genes' may actually target only a subset of isoforms of the gene, and it is critically important to know which cells express these isoforms to predict on- and off-target drug interactions.

Genome-wide characterization of cell-type-specific splicing is still lacking mainly due to inherent challenges in scRNA-seq such as data sparsity. The field still debates whether single-cell splicing heterogeneity constitutes another layer of splicing regulation or is dominated by stochastic but stereotyped 'binary' exon inclusion (*Arzalluz-Luque and Conesa, 2018*; *Buen Abad Najar et al., 2020*), and whether cells' spliced RNA is sequenced deeply enough in scRNA-seq for biologically meaningful inference. Most differential splicing analysis requires isoform estimation, which is unreliable with low or biased counts (*Westoby et al., 2020*), or 'percent spliced in' (PSI) point estimates, which suffer from high variance at low read depth and amplify the multiple hypothesis testing problem (*Arzalluz-Luque and Conesa, 2018*; *Buen Abad Najar et al., 2020*). Most methods for splicing analysis from scRNA-seq data are not designed for droplet-based data (*Huang and Sanguinetti, 2017*; *Song et al., 2017*). Studies of splicing in scRNA-seq data have mostly focused on just a single cell type or organ and used pseudo-bulked data before differential splicing is analyzed, thus do not provide the potential to discover new subclusters or provide bona fide quantification of splicing at single-cell resolution. Further, studies have almost exclusively used full-length data such as Smart-seq2 (SS2) (*Arzalluz-Luque and Conesa, 2018*; *Buen Abad Najar et al., 2020*). Without genome-wide resolution, global splicing trends are missed and the focus on full-length sequencing data means that single cells sequenced with droplet-based technology, the majority of sequenced single cells including many cell types that are not captured by SS2 (*Svensson et al., 2020*; *Travaglini et al., 2020*), have been neglected (*Patrick et al., 2020*).

To overcome statistical challenges that have prevented analysis of cell-type-specific alternative splicing, we used the SpliZ (*Olivieri et al., 2021*), a statistical approach that generalizes PSI (*Figure 1A*, Materials and methods) and increases the power to detect cell-type-specific alternative splicing in single cells. As detailed in *Olivieri et al., 2021*, for each gene, the SpliZ quantifies splicing deviation in each cell from the population average. A large negative (resp. positive) SpliZ score for a gene in a cell means that the cell has shorter (resp. longer) introns than the average for that gene. Highlighting its disciplined statistical nature, the SpliZ reduces to PSI in the simplest exon skipping case (*Olivieri et al., 2021*).

When cell type annotations are available, the SpliZ statistically identifies genes with cell-type-specific splicing patterns. The SpliZ is an unbiased and annotation-free algorithm and is applicable to both droplet-based and full-length scRNA-seq technologies. The SpliZ attains high power to detect differential alternative splicing in scRNA-seq when genes are variably and sparsely sampled (*Figure 1B*) by controlling for sparsely sampled counts and technological biases such as those introduced by 10X Chromium (10X). Because the SpliZ gives a single value for each gene and each cell, it enables analyses beyond differential splicing between cell types, including correlation of splicing changes with developmental trajectories and subcluster discovery within cell types based on splicing differences (*Figure 1C and D*). It also provides a statistical, completely annotation-free approach that identifies splice sites called SpliZsites that contribute most variation to cell-type-specific splicing as measured by analysis of SpliZ components through the singular value decomposition (*Olivieri et al., 2021*).

Here, we used the SpliZ to analyze 75,789 cells profiled with 10X across 12 tissues and 82 cell types from one human individual through the *Tabula Sapiens* project (*Tabula Sapiens Consortium, 2021*). We also performed SpliZ analysis on a second human and two mouse lemur and two mouse individuals: together we analyzed 109,981 human (*Tabula Sapiens Consortium, 2021*), 165,200 mouse

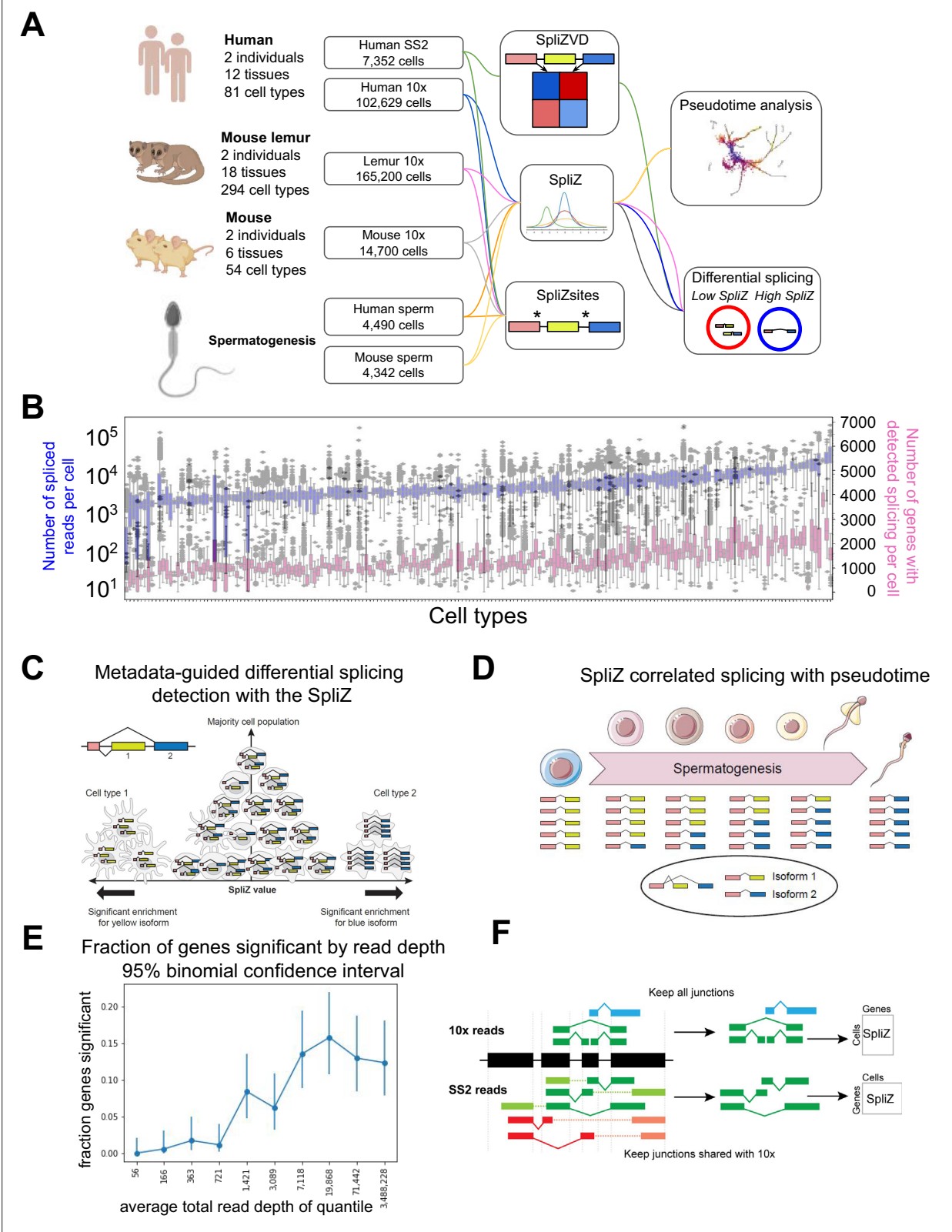

**Figure 1.** Analysis of alternative splicing in single-cell RNA-seq. (**A**) Human, mouse lemur, and mouse single-cell RNA-seq from 10X and SS2 were run through the SpliZ pipeline for differential splicing discovery. (**B**) 10X data from the first human individual contains 82 cell types with variable sequencing depth. (**C**) Given cell type annotation, SpliZ scores can be aggregated for each cell type, allowing identification of cell types with statistically deviant splicing. (**D**) Cell-wise SpliZ values can be correlated with pseudotime to identify developmentally regulated alternative splicing. (**E**) The fraction of

*Figure 1 continued on next page*

*Figure 1 continued*

genes called as having significant differential alternative splicing by cell type is higher at higher sequencing depths, plateauing at around 20,000 spliced reads in the dataset, at which point around 15% of genes were called as significant. (**F**) The SpliZ is calculated independently for SS2 data restricted to junctions found in 10X data, and used to validate results from 10X data.

lemur (*Tabula Microcebus Consortium, 2021*), and 14,700 mouse cells (*Tabula Muris Consortium, 2018*) sequenced with 10X (*Figure 1A*, *Supplementary file 1*). Additionally, we analyzed spermatogenesis trajectories across 4490, 4342, and 5601 10 X sperm cells from human (*Hermann et al., 2018*), mouse (*Hermann et al., 2018*), and mouse lemur (*Tabula Microcebus Consortium, 2021*), respectively. The SpliZ has higher power to detect differential alternative splicing between cell types at higher sequencing depths, plateauing at around 20,000 spliced reads measured for the gene, at which point around 15% of genes were called as significant (*Figure 1E*).

We performed high-throughput computational validation with the SS2 cells (*Figure 1F*) along with experimental and in situ validations including Sanger sequencing and RNA FISH on cells from the lung and muscle. Mouse and mouse lemur data was used to assess evolutionary conservation of the discoveries in human. Examples found by this analysis include differential cell-type-specific and compartment-specific alternative splicing in a subset of ubiquitously expressed genes including *MYL6*, an actin light chain subunit, *RPS24*, a core ribosomal subunit associated with Diamond-Blackfan Anemia (*Gupta and Warner, 2014*), and *TPM1*, a tumor suppressor tropomyosin. Knockout studies of RNA-binding proteins have implied the importance of alternative splicing in spermatogenesis; however, comprehensive profiling of alternative splicing in normal spermatogenesis has not been possible. In this study, for the first time we identify regulated splicing changes in 170 genes during normal human spermatogenesis using rigorous statistical methodology for automatic computational single-cell splicing profiling, including conserved regulated splicing in centrosomal protein domain and lncRNA.

To our knowledge, this work provides the first unbiased and systematic screen for cell-type-specific splicing regulation in highly resolved human cells, predicting functionally significant alternative splicing, and calls for more attention to the potential of scRNA-seq for discovering regulated splicing in single cells.

## Results

### Conserved splicing in ubiquitously expressed genes, including *ATP5FC1* and *RPS24*, predicts cellular compartment at single-cell resolution

We applied the SpliZ (*Olivieri et al., 2021*), a recently developed method to identify cell-type-specific splicing, to ~75k 10X cells in 12 tissues from one human donor (*Tabula Sapiens Consortium, 2021*), beginning by testing for splicing regulation differing by tissue compartment (immune, epithelial, endothelial, and stromal) regardless of the tissue of origin (SpliZ scores available for download at the following FigShare repository: DOI: 10.6084 /m9.figshare.14531721). This analysis identified 1.6% (22 of 1353) of genes with computable SpliZ scores as having consistent compartment-specific splicing effects (*Supplementary file 2*, Materials and methods). *ATP5F1C*, *RPS24*, and *MYL6* have the highest compartment-specific splicing effects, defined as the largest magnitude median SpliZ in any compartment, and their compartment-specific splicing was conserved in mouse and mouse lemur. *ATP5F1C* is the gamma subunit of mitochondrial ATP synthase, a multi-subunit molecular motor that converts the energy of the proton potential across the mitochondrial membrane into ATP. *MYL6* is an actin light chain subunit known to have cell-type-specific splicing differences in the muscle (*Brozovich et al., 2016*). *RPS24* is an essential ribosomal protein for ribosome small subunit 40S discussed in detail later. Among the examples of genes demonstrating compartment-specific splicing is *LIMCH1* (*Figure 2A*). *LIMCH1* has been reported as a non-muscle myosin regulator (*Lin et al., 2017*) and has been associated with Huntington's disease (*Lin et al., 2016*) with little other characterization, including, to our knowledge, no reports of regulated splicing. The SpliZ values for *MYL6*, *RPS24*, and *ATP5F1C* are not correlated with gene expression (*Figure 2—figure supplements 1–3*).

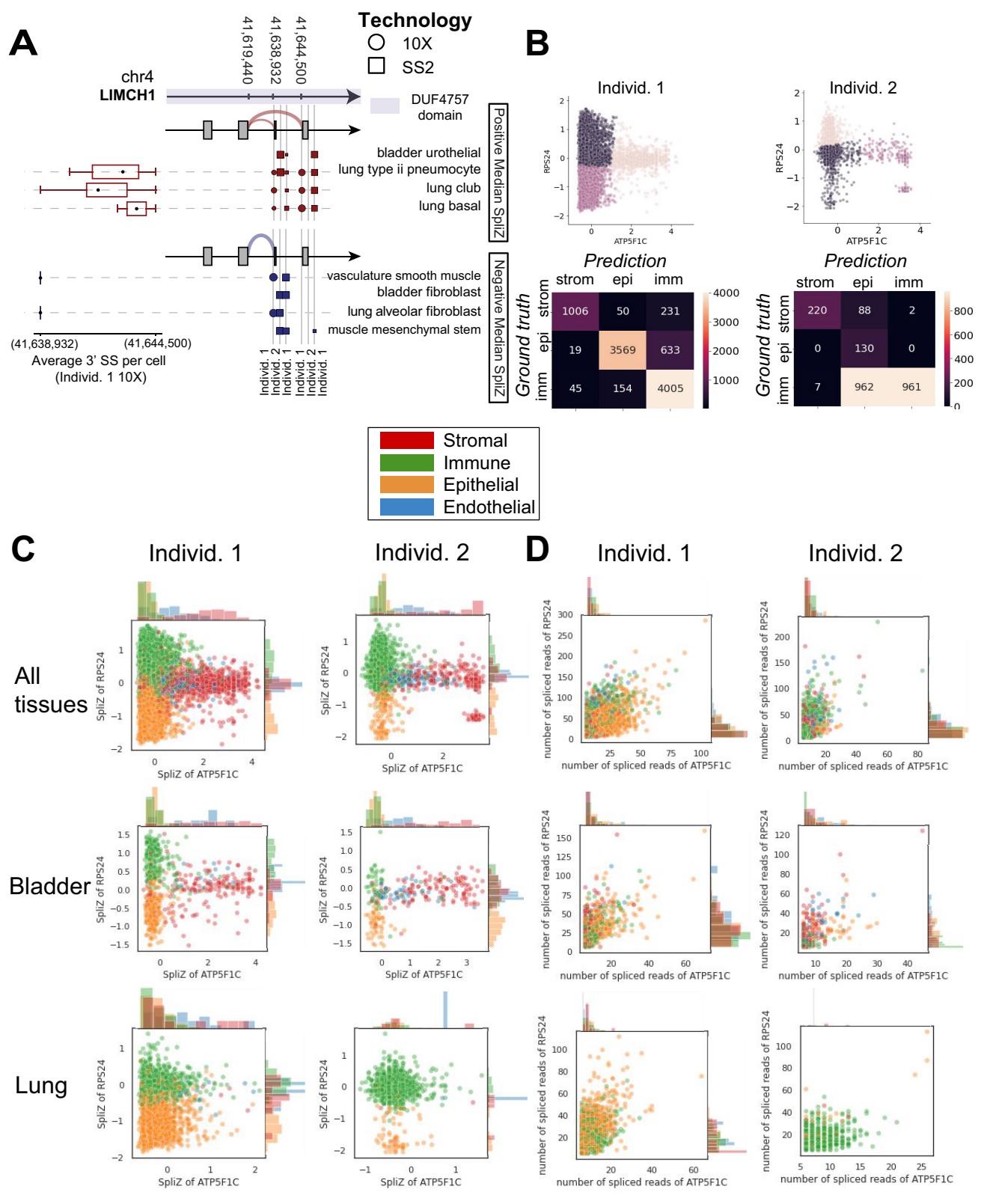

**Figure 2.** Compartment-specific alternative splicing revealed by applying the SpliZ to scRNA-seq data. (**A**) Dot and sashimi plots showing *LIMCH1* compartment-specific exon skipping (involving 5' splice site [5' SS] 41,619,440 and two 3' splice sites [3' SS] 41,638,932 and 41,644,500) impacting a protein domain of unknown function DUF4757 (shown by the purple color on the gene structure) across cell types and 10X and SS2 data from both human individuals. Each dot shows junction expression for the splice junction from the 5' SS to one of the 3' SS's, with dot size proportional to the

*Figure 2 continued on next page*

*Figure 2 continued*

fraction of junctional reads supporting the splice junction in that cell type and dataset. Columns of dots are biological replicates; the first column is the individual 1 10X dataset (circles) and the next two columns are SS2 datasets from individuals 1 and 2 (squares). Cell types are grouped in two sets depending on the sign of the median SpliZ score in 10X data from human individual 1. The thickness of the sashimi arcs represents the fraction of the reads mapping to each 3′ SS when all datasets and corresponding cell types for the sashimi arc are grouped together. The box plot for each cell type shows the distribution of the weighted average 3′ SS (weights being the number of reads aligning to each 3′ SS in the cell) for each cell and the reads are assigned 1 (for those aligning to the closer 3′ SS) and 2 (for those aligning to the farther 3′ SS). Stromal cells including vasculature smooth muscle cells and fibroblasts always include the exon (higher fraction of reads aligning to the splice site at 41,638,932), while epithelial cells including bladder urothelial cells skip with >50% rate. (**B**) Unsupervised k means clustering results in 78, 84, and 95% accuracy of compartment classification for the stromal, epithelial, and immune compartments, respectively, for individual 1, and 70, 100, and 49% , respectively, for individual 2. (**C**) The SpliZ scores of the genes *ATP5F1C* and *RPS24* together separate compartments in both human individuals. Each dot represents the SpliZ score in a single cell and is color coded by the compartment. (**D**) Using the spliced read counts for each gene rather than the SpliZ does not separate the compartments, showing that this separation is not driven by gene expression differences. Each dot represents the number of spliced reads in a single cell and is color coded by the compartment.

The online version of this article includes the following figure supplement(s) for figure 2:

**Figure supplement 1.** Gene expression and the SpliZ value of *MYL6* across single cells in the *Tabula Sapiens* dataset.

**Figure supplement 2.** Gene expression and the SpliZ value of *RPS24* across single cells in the *Tabula Sapiens* dataset.

**Figure supplement 3.** Gene expression and the SpliZ value of *ATP5F1C* across single cells in the *Tabula Sapiens* dataset.

To test the predictive power of compartment-specific genes at single-cell resolution, we performed unsupervised k-means clustering on the SpliZ scores of *RPS24* and *ATP5F1C* alone. Setting $k = 3$, cells from stromal, epithelial, and immune compartments in the first human individual were classified with accuracies of 78, 84, and 95%, respectively, independent of gene expression (70, 100, and 49% in the second individual) (*Figure 2B–D*, Materials and methods). The lower accuracy for individual 2 may be caused by individual 2 having only a third as many cells. The endothelial compartment was not included because it had a small proportion of cells in both datasets (3.7% in individual 1, 4.5% in individual 2). This establishes that splicing of a minimal set of genes, in this case only two, has high predictive power of the compartmental origin of each single cell. Underlining tight biological regulation of splicing in these genes, parallel analysis in the 10X scRNA-seq data from mouse lemur and mouse shows compartment-specific splicing is conserved for *RPS24* and *MYL6* (*Figures 3 and 4*).

## The splicing of actin regulator *MYL6* is compartment-specifically regulated

We identified *MYL6* as both cell-type-specifically and compartment-specifically spliced in humans and its splicing pattern is conserved (*Figure 3*). *MYL6* is a ubiquitously expressed myosin light chain subunit and is known to have a lower level of exon skipping in muscle than non-muscle tissue (*Brozovich et al., 2016*), but differential exon skipping at a single-cell level has only been characterized in smooth muscle cells. We find in human, mouse, and mouse lemur that the stromal compartment, which includes smooth muscle, as well as the endothelial compartment have a relatively higher proportion of exon inclusion, while the epithelial compartment has a lower level of exon inclusion and the immune compartment has the lowest level of exon inclusion (*Figure 3A–E*, *Figure 3—figure supplement 1*). Despite these trends being the same for all three species, mouse has higher levels of exon inclusion in all compartments than in the other two species.

We validated compartment-specific differential alternative splicing in *MYL6* using RNA FISH with isoform-specific probes on human adult lung tissue obtained from the Stanford Tissue Bank (*Figure 3F*, Materials and methods). In human lung, this confirmed that bronchiole smooth muscle cells have the highest fraction of the exon inclusion isoform (57%), while the respiratory epithelium has a lower fraction of this isoform (16%) and the two profiled immune cell types (macrophages and lymphocytes) have the lowest fractions of the exon inclusion isoform (10% and 8%, respectively). We separately performed RNA FISH on human cells isolated from the muscle, which showed that mesenchymal stem cells and muscle stem cells have a higher proportion of the exon inclusion isoform than endothelial cells (*Figure 3—figure supplement 2*, Materials and methods).

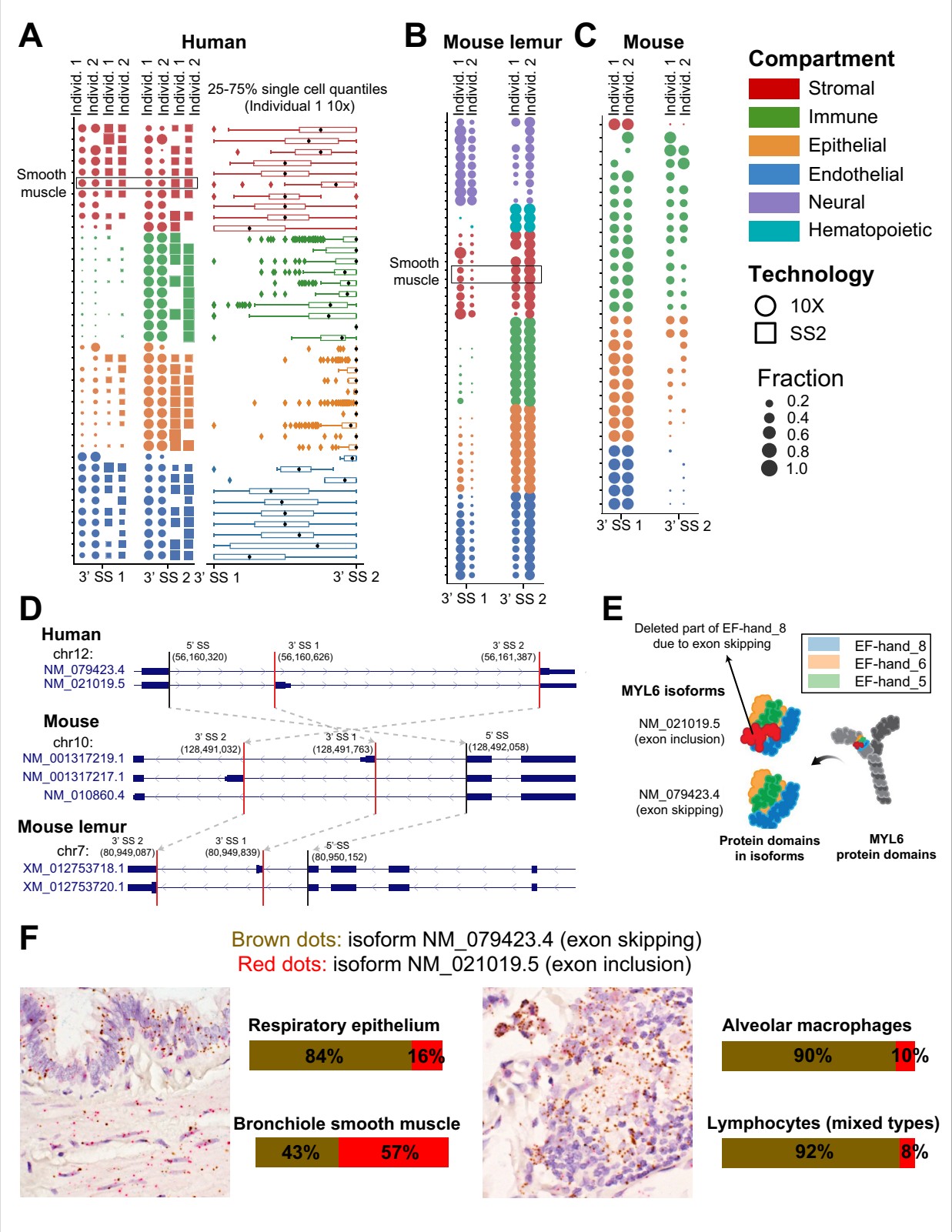

**Figure 3.** Compartment-specific alternative splicing in *MYL6*. Differential alternative splicing between compartments for *MYL6* is driven by an exon skipping event with orthologous splice sites (SS) in (**A**) human (5' SS: 56,160,320 and two 3' SSs: 56,161,387 and 56,160,626), (**B**) mouse lemur, and (**C**) mouse. Each dot shows the expression for the splicing to one of the 3' SSs marked by vertical red lines on the gene annotation in (**D**) in a 10X (circles) or SS2 (squares) dataset from individuals 1 and 2. Columns of dots are biological replicates; for human data, the first two columns are 10X and the second

*Figure 3 continued on next page*

*Figure 3 continued*

two columns are SS2. Dots are colored by compartment. For mouse and mouse lemur, the two columns are 10X samples. The box plot is obtained by assigning 1 and 2 to the closer and farther 3' SS and then computing their weighted average for each cell according to their corresponding fraction of junctional reads in the cell. Cells in the immune compartment have higher exon skipping rates than cells in the stromal compartment in all three organisms. Smooth muscle cell types are boxed within the stromal compartment. Mouse cells have the same relative proportions of exon inclusion between compartments, but express higher levels of the exon included isoform overall. The SpliZ scores (and also gene expression values) for *MYL6* across all 10X cells in human individual 1 are shown in *Figure 2—figure supplement 1*. (**D**) Gene structures showing *MYL6* annotation in human, mouse, and mouse lemur. The gray arrows between different organisms show LiftOver mapping between human, mouse, and mouse lemur, indicating that orthologous splice sites are involved in alternative splicing in different organisms. (**E**) Protein domains in MYL6 and how they are organized in the two *MYL6* isoforms. The exon skipping leads to the deletion in the EF_hand_8 domain (shown by the red color). (**F**) RNA FISH validation in human lung: each slide is stained simultaneously with probes in red (specific to exon inclusion) and brown (specific to exon exclusion). As found from the scRNA-seq data, smooth muscle cells have a higher proportion of the included exon than the other compartments and immune cells have the lowest proportion.

The online version of this article includes the following figure supplement(s) for figure 3:

**Figure supplement 1.** Uncropped plots of *MYL6* expression.

**Figure supplement 2.** FISH validation for *MYL6* alternative splicing in cells isolated from human muscle.

## *RPS24* has compartmentally regulated alternative splicing and expresses a microexon in human epithelial cells

*RPS24* is a highly expressed and essential ribosomal protein. Our analysis revealed that *RPS24* has the most significant cell-type-specific and compartment-specific alternative splicing patterns at its C terminus in human and mouse lemur. The significance of the alternative splicing patterns of *RPS24* is underscored by recent findings that ribosome composition is more modular than previously appreciated in a cell- and tissue-specific manner (**Genuth and Barna, 2018**). There has been a partial study of *RPS24* splicing treating two isoforms (**Song et al., 2017**), and another study reported modest differential splicing at the tissue level (**Gupta and Warner, 2014**; **Song et al., 2017**) for *RPS24* involving three isoforms. However, here, we show that splicing of *RPS24* is more complex and highly regulated at a single-cell level (**Figure 4A and B**).

Differential alternative splicing of *RPS24* involves alternate inclusion of three short exons, *a*, *b*, and *c*, each only 3, 18, and 22 nucleotides long, respectively (**Figure 4C**); regions of genomic sequence around exon *a* are ultraconserved. Splicing of the *a*, *b*, and *c* exons results in isoforms whose protein domains differ by the presence of a single lysine at the solvent-exposed site of the ribosome, and some isoforms (e.g., +*a-b+c* and -*a+b+ c*) have no change of amino acids. The -*a-b+c* isoform is dominant in all endothelial cell types in human, as well as most stromal cell types and half of the immune cell types (**Figure 4A**). Within the immune compartment, our global analysis reveals differential alternative splicing of *RPS24* in monocytes, where the -*a-b-c* isoform is dominant in classical monocytes residing in multiple tissues and the -*a-b+c* isoform is dominant in non-classical monocytes. Intermediate monocytes have equal proportions of each.

Epithelial cell types in human are marked by the dominance of the +*a-b+c* isoform (as shown by the fraction of the pink splice junction in **Figure 4A**), which is barely present in any non-epithelial cell types and is not dominant in any of them, and only differs by three nucleotides from the -*a-b+c* isoform. The +*a-b+c* isoform is only found at very low levels in mouse and mouse lemur. The human epithelial specificity of the +*a-b+c* isoform is further supported by single-cell RT-PCR (**Figure 4D**).

Other cell types have distinct isoform expression as well: the -*a+b+ c* isoform is specific to fast and smooth muscle cells in both human and mouse lemur, such as thymus fast muscle and bladder smooth muscle in human, as well as vascular-associated bladder smooth muscle in mouse lemur, though some smooth muscle cell types in human do not express it, specifically thymus vascular-associated smooth muscle and vasculature smooth muscle. Among profiled cell types, the +*a+b+c* isoform is found only in neural retinal cells in the mouse lemur, the only dominant isoform including the microexon *a* in the mouse lemur (retina data not available for human).

In addition to using RNA FISH to independently validate cell-type-specific splicing in a subset of lung and muscle cells (**Figure 4E**, **Figure 4—figure supplement 1**), we performed high-throughput validation using Bowtie2 alignment (**Langmead and Salzberg, 2012**; **Figure 4—figure supplement 2**). The Bowtie2 alignment data confirms that the +*a-b+c* isoform is present at low levels in the epithelium of mouse and mouse lemur compared to high levels in human epithelium. The RNA FISH data

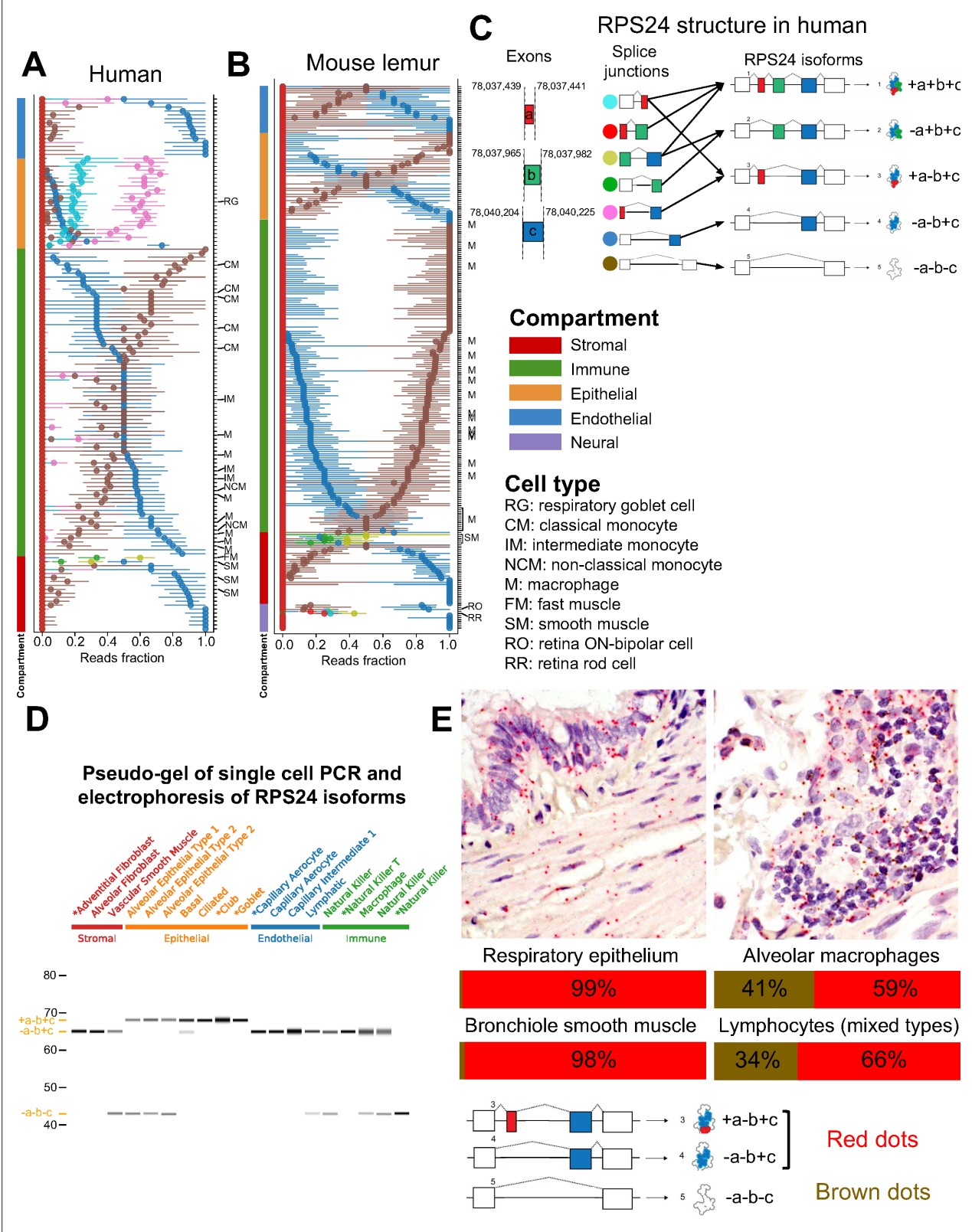

**Figure 4.** RPS24 has striking compartment-specific alternative splicing. (**A**) Each colored circle in the plot represents one *RPS24* junction that uniquely identifies an isoform (junction with green circle represents two isoforms). For each cell type (y axis), the median of all single-cell point estimates of junction fraction in the cell type is plotted on the x axis, with bars representing the 25th and 75th quantiles of single-cell junction fractions. Cell types are sorted by compartment. Black arrows show the splice junctions that can be used for identifying each isoform. Within the immune compartment, the

*Figure 4 continued on next page*

*Figure 4 continued*

fraction of the blue junction increases from classical monocytes to intermediate monocytes to non-classical monocytes. (**B**) The isoform with epithelial-specific splicing in human is not expressed in mouse lemur. However, the same isoform is expressed in smooth muscle as in human. Retinal cells are the only cells to express the +*a*+*b*+*c* isoform. (**C**) *RPS24* isoform structure in human shows alternative inclusion of three cassette exons *a*, *b*, and *c* create five annotated isoforms. (**D**) Single-cell PCR validates the prediction that the +*a*-*b*-*c* isoform is epithelial-specific. All the epithelial cells contain the isoform with the 3-nt exon *a* , while none of the cells from other compartments do. PCR products from the cells prefixed by asterisks were Sanger-sequenced and matched the expected isoform without evidence of mixture. (**E**) *RPS24* FISH in human lung validates scRNA-seq computational predictions. Slides were simultaneously stained with probes in red and brown, specific for alternative splice junctions. As found from the scRNA-seq data, respiratory epithelium and bronchiole smooth muscle in the epithelial and stromal compartments, respectively, have a low proportion of the -*a*-*b*-*c* isoform compared to alveolar macrophages and lymphocytes, both of which are in the immune compartment.

The online version of this article includes the following figure supplement(s) for figure 4:

**Figure supplement 1.** FISH validation for *RPS24* alternative splicing in cells isolated from human muscle.

**Figure supplement 2.** *RPS24* isoforms quantified by the Bowtie2 aligner validate STAR-based discoveries.

confirms that the -*a*-*b*-*c* isoform composes just ~1% to 2% in the respiratory epithelium and bronchiole smooth muscle, while alveolar macrophages and lymphocytes have about 34–41% -*a*-*b*-*c* (*Figure 4E*).

Together, the subtle changes in protein sequence from alternative splicing of *RPS24* prompt two hypotheses: one is that splicing affects post-translational modifications (*Kondrashov et al., 2011*). However, the fact that some splice variants have subtle or no variation in the encoded protein suggests an alternative that, like isoforms of *Actin* in mouse, *RPS24* splicing could function at the nucleotide rather than protein level (*Vedula et al., 2017*).

## Approximately 9% of measured genes have cell-type-specific splicing regulation

Splicing regulation in the vast majority of human cell types has not been characterized. We used the 82 annotated cell types in the *Tabula Sapiens* cell atlas to identify genes with statistical support for having differential alternative splicing patterns using the same SpliZ procedure for identifying compartment-specific genes (*Olivieri et al., 2021*; *Figure 1C*, Materials and methods, *Supplementary file 3*). Among genes called significant, the Pearson correlation between the median SpliZ in individuals 1 and 2 (10X) was 0.77 and it was 0.44 between 10X and SS2 within individual 1 (p-value < 10e-50, Materials and methods, *Figure 5*). 129 out of 1416 genes (9%) had significant cell-type-specific splicing profiles based on discovery with 10X data from individual 1 (p-value < 0.05, effect size >0.5) (Materials and methods, *Figure 5—figure supplement 1*). Genes with cell-type-specific splicing regulation include *TPM1* (*Figure 6*), *PNRC1* (*Figure 7A*), and *FYB* (*Figure 7—figure supplement 1*), among others.

Tropomyosin 1, *TPM1,* which has three isoforms each impacting the tropomyosin domain at the 3′ end of the transcript, served as a positive control in this analysis as it is known to undergo cell-type-specific splicing. It is ranked as the 27th most significant effect size. Unbiased SpliZ analysis finds that capillary endothelial cells express about equal levels of three isoforms, while smooth muscle cells almost exclusively express the isoform with the 3′-most domain (*Figure 6A–C*). This trend, among others, is consistent with knowledge of *TPM1* splicing from other studies (*Gooding and Smith, 2008*), though it extends its splicing profile to cell types where it has never been characterized. Among the comprehensive catalog of differences, smooth muscle and pericyte cells consistently include different cassette exons at the 3′ end of the transcript, affecting the tropomyosin protein domain (*Figure 6A*). Cell types outside of muscle such as bladder pericytes and bladder fibroblasts have similar splicing profiles as smooth muscle and muscle mesenchymal stem cells, respectively. Splicing biology of *TPM2* and *TPM3,* two other genes from the *TPM* family where partial characterization has suggested cell-type-specific splicing, is similarly rediscovered in our analysis and significantly extended: slow muscle cells (and fast muscle cells for *TPM2*) have different splicing patterns than other cell types for both genes (*Figure 6—figure supplement 1*).

*PNRC1*, a nuclear receptor coregulator that functions as a tumor suppressor, has the fifth highest effect size (*Figure 7A*; *Gaviraghi et al., 2018*); limited in vitro studies have found evidence that splice variants of *PNRC1* modify its interaction domains and nuclear functions (*Wang et al., 2008*). The largest magnitude median SpliZ score for *PNRC1* is found in muscle stromal mesenchymal stem

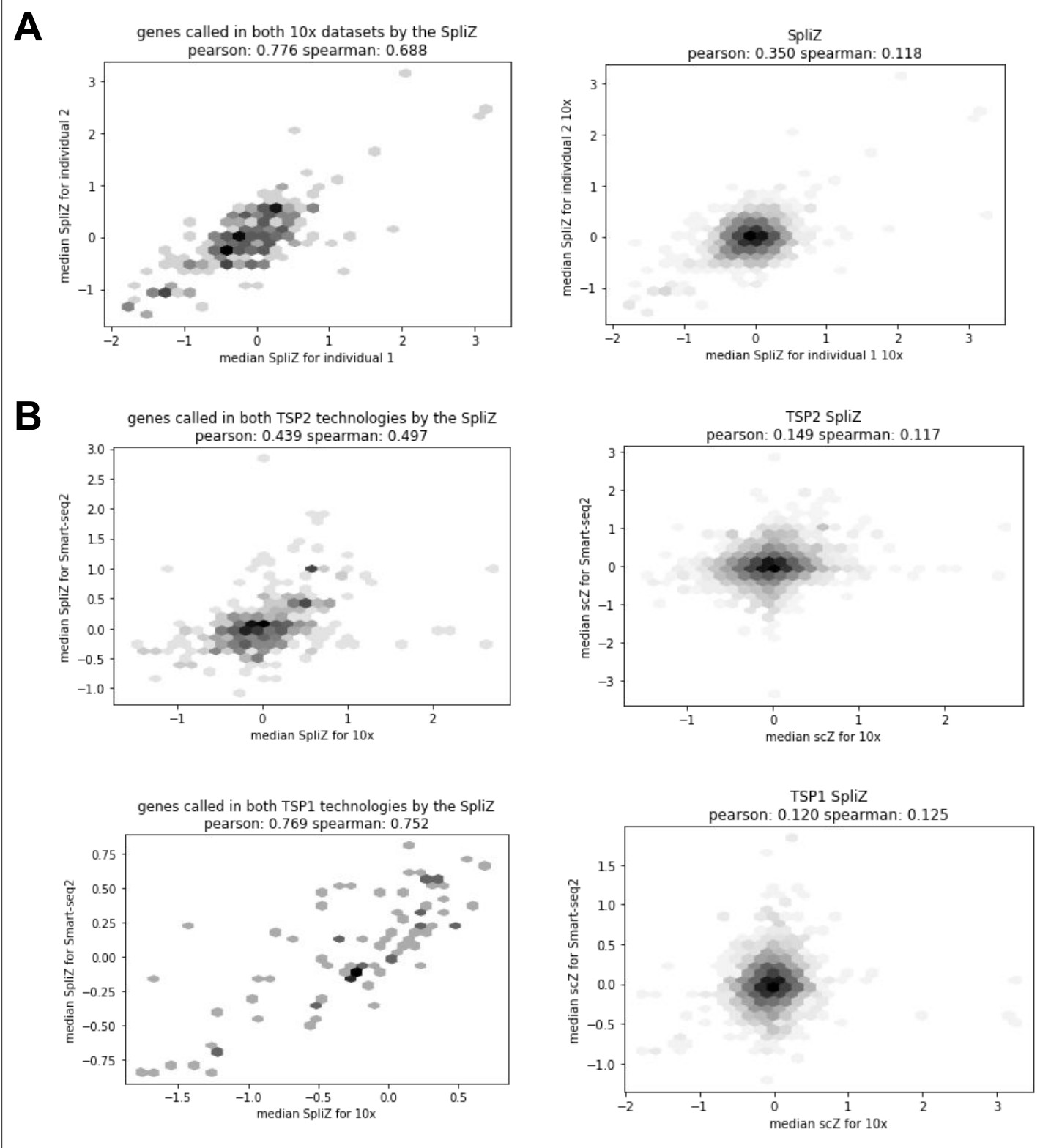

**Figure 5.** Correlations show high concordance of the SpliZ values for significant genes across biological replicates. (**A**) When subsetted to only shared junctions and shared cell types, the SpliZ values for significant genes for both 10X datasets are highly concordant (Pearson correlation of 0.776). (**B**) Comparing datasets from the 10X and SS2 technologies.

The online version of this article includes the following figure supplement(s) for figure 5:

**Figure supplement 1.** Choosing effect size filters.

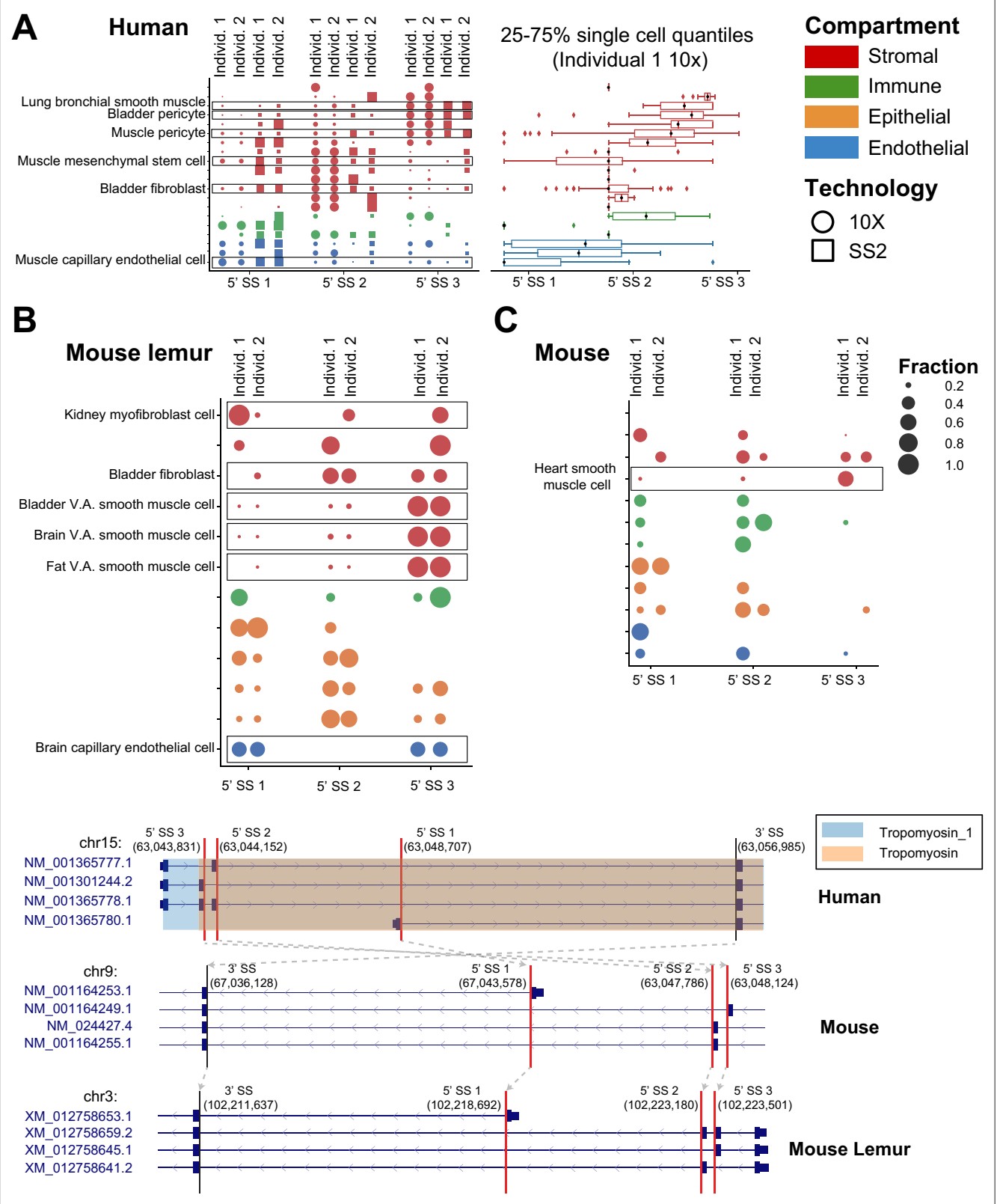

**Figure 6.** Cell-type-specific and conserved alternative splicing in *TPM1*. Conserved splicing in *TPM1* is recovered in (**A**) human, (**B**) mouse lemur, and (**C**) mouse. *TPM1* has a pattern of differential splicing involving two cassette exons and an alternate 5′ end (as shown by the gene structures at the bottom of the figure). Capillary endothelial cells mostly express the isoform with the alternate 5′ end (5′ splice site [5′ SS] 1), while smooth muscle almost exclusively expresses the isoform with the 5′-most domain (boxed in the figure). The box plot shows the distribution of the average 5′ SS (obtained as

*Figure 6 continued on next page*

*Figure 6 continued*

the weighted average of 5' SS when ranked from 1 to 3 from the closest to the farthest according to their fraction of junctional reads) for the cells within a cell type (see *Figures 2 and 3* for more explanation of dot and box plots). There is differential isoform usage within the stromal compartment as well, for example, human bladder stromal fibroblasts and bladder stromal pericytes each express a different dominant cassette exon. Both lemur and mouse similarly express cell-type-specific differences in *TPM1* isoform usage. Orthologous SpliZsites in human, mouse, and mouse lemur are involved in alternative splicing based on the LiftOver mapping, as shown by gray arrows on the gene structures.

The online version of this article includes the following figure supplement(s) for figure 6:

**Figure supplement 1.** Cell-type-specific splicing in other members of the tropomyosin family.

cells, revealing new splicing regulation. Other cell types, including immune and stromal types in the bladder, have markedly distinct splicing programs (*Figure 7A*).

The high dimensionality of SpliZ scores enabled us to test if unsupervised clustering on the median SpliZ scores could recapitulate relationships between cell types (*Figure 7B*). We found that the same cell types from different tissues are generally clustered together, including macrophages and T cells from different tissues and also intestinal cell types from large and small intestine (*Figure 7B*, Materials and methods). This clustering also reveals that the splicing programs of cell types from the same compartment are highly similar and automatically clustered together independent of their tissues (*Figure 7B*).

## The most statistically variable splice sites with cell-type-specific regulated splicing are annotated splice sites involved in unannotated alternative splicing

The biological importance of splicing detected by the SpliZ and the fact that it is completely agnostic to isoform annotation led us to test whether cell-type-specific splicing variation is (a) focused at exons that are annotated to undergo alternative splicing and (b) conserved. The SpliZ method uses a statistical, annotation-free approach to identify SpliZsites: variable splice sites that contribute most to the cell-type-specific splicing of a gene agnostic to gene annotation. SpliZsites blindly reidentify known alternative splice sites in *ATP5F1C*, *MYL6*, *TPM1*, and *RPS24* (*Supplementary file 4*). In *TPM2*, the SpliZ reidentifies two known alternative splicing sites but also predicts a cell-type-specific unannotated alternative splicing event in stromal cell types involving 5' splice site 35,684,315 affecting Tropomyosin and Tropomyosin 1 protein domains.

Genome-wide, the vast majority of SpliZsites (93%) in significant genes are at boundaries of annotated exons. However, only 38.5% are annotated as alternatively spliced, suggesting that unannotated – rather than annotated – exon skipping accounts for underappreciated splicing variation in single cells. Further, exon skipping has a global effect on single-cell proteomes: more than half of SpliZsites impact protein coding domains; 34% impact the 3' UTR and 16% impact the 5' UTR, consistent with a bias in 10X technology towards the 3' gene end. Supporting the idea that SpliZsites discover a real biological signal, 15.5% of LiftOver human SpliZsites were also SpliZsites in the mouse lemur compared to 7% expected under the null (Materials and methods). Only 8.0% of LiftOver SpliZsites in human were called as SpliZsites in mouse compared to 8.8% expected under the null. This could be due to many factors including a larger evolutionary distance between mouse and human, smaller number of analyzed mouse cells, or lower sequencing depth (*Supplementary file 1*).

## The SpliZ identifies subpopulations of classical monocytes with distinct splicing of an ultraconserved exon of *SAT1*

The SpliZ has a theoretical normal distribution under the assumption that cells within a cell type all have the same propensity to express each splice isoform (the 'null hypothesis'). This property allows us to statistically test whether cell types subcluster on the basis of splice isoform, as quantified by the SpliZ, using an integrated complete-data likelihood (ICL) model selection framework. This is based on Gaussian mixture modeling (GMM), and includes a measure of 'effect size' differences between clusters via the Bhattacharyya distance, a measure of the distance between probability distributions (Materials and methods). Importantly, this approach avoids false-positive calls of apparent binary exon inclusion (*Buen Abad Najar et al., 2020*; Materials and methods, manuscript in preparation). We applied the ICL analysis of the SpliZ to immune cell types in individual 2 to illustrate the power of

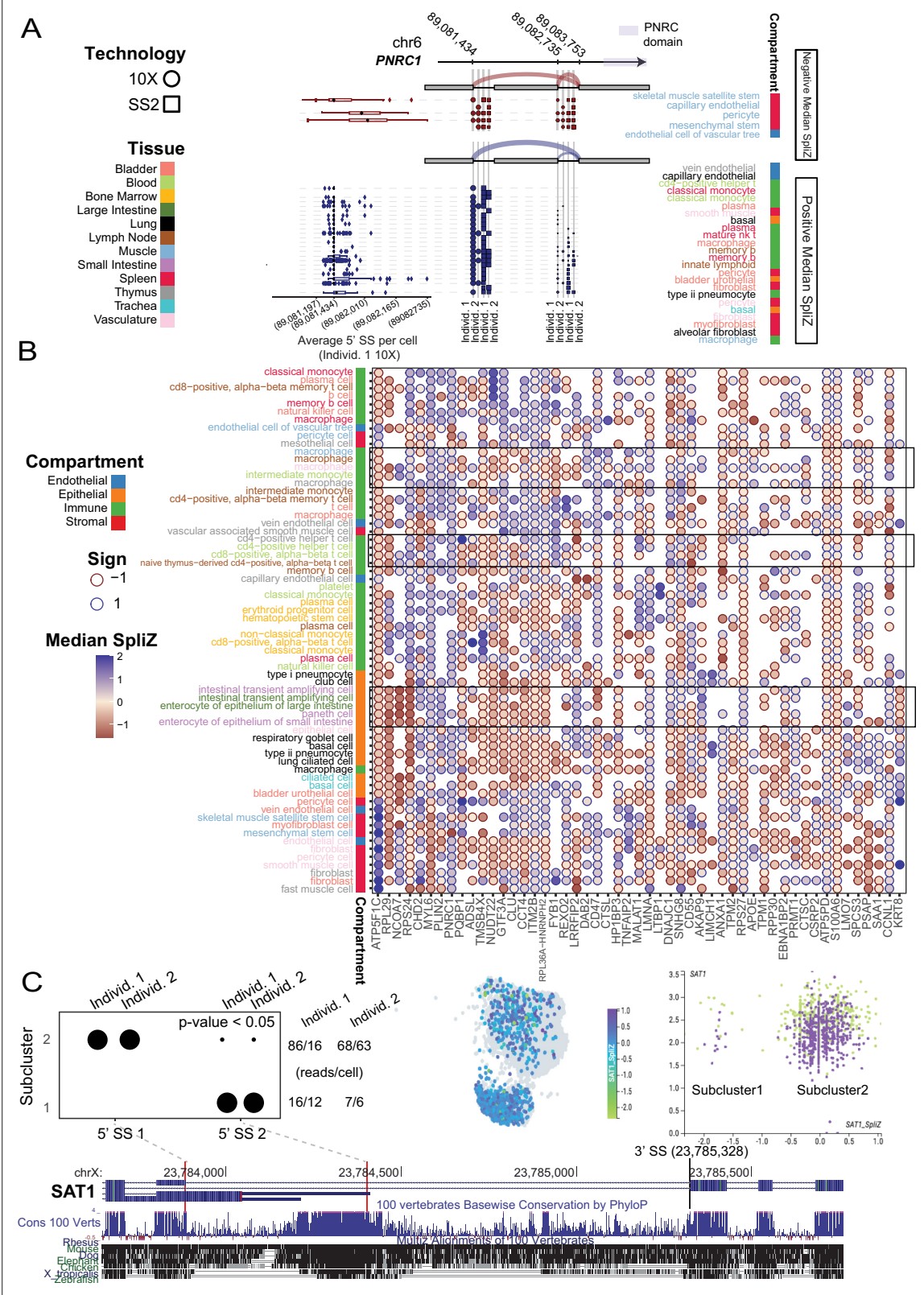

**Figure 7.** Cell-type-specific alternative splicing is prevalent in human genes and can reveal novel cell subpopulations. (**A**) Cell-type-specific exon skipping in *PNRC1* (involving one 3' SS and two 5' SS's) is replicated across the four human datasets. Skeletal muscle satellite stem cells include the exon about 50% of the time, whereas vein endothelial cells in the thymus never include the exon. Cell types with negative median SpliZ values are on the top panel, and those with positive median SpliZ values are on the bottom panel. Each of the four human datasets is plotted, with circles representing

*Figure 7 continued on next page*

*Figure 7 continued*

10X data and squares representing SS2 data. The gene annotation is shown above the dots, with sashimi arcs indicating the mean expression of each junction for the given cell types and datasets in the corresponding panel (similar to *Figure 2*). The known protein domain is marked on the gene structure. Box plots for each cell type are based on weighted 3' usage (based on the fraction of junctional reads to each 3' SS) of the 5' SS for each cell in the cell type in individual 1 10X data (**L**). Each box shows 25–75% quantiles of average 5' SS per cell. (**B**) Unsupervised clustering analysis with the SpliZ identified clusters of cell types and compartments independent of tissue. Dots show the median SpliZ (effect size) for genes found to be significantly regulated across cell types. Only 50 significant genes with the highest effect size and cell types with >25 significant genes are shown. Hierarchical clustering was performed on both genes and cell types based on median SpliZ values. Cell type names are color-coded based on their tissue (same tissue colors as in **A**) and the side bar shows the compartment for each cell type. (**C**) Alternative splicing of gene *SAT1* distinguishes two populations of cells within blood classical monocytes and involves an ultraconserved exon. The dot plot shows the differential inclusion of the 5' SSs for the 3' SS at 23,785,300 for cells grouped based on their assigned subclusters. The number of reads (X) and cells (Y) containing the splice junctions involving the 3' SS in each individual are shown at right. Clustering based on gene expression as shown by cellxgene visualization (middle panel) and scatter plot (right panel) does not distinguish cell populations with distinct splice profiles. In the scatter plot, the x and y axes represent the gene expression and SpliZ values for *SAT1* in each cell, respectively. Cells are colored according to their human individual number. Visualization of the gene expression value for *SAT1* does not distinguish the populations; both subclusters contain classical monocytes from both human individuals (right scatter plot).

The online version of this article includes the following figure supplement(s) for figure 7:

**Figure supplement 1.** Differential alternative splicing of *FYB1* within the immune compartment.

single-cell splicing quantification by the SpliZ for defining subsets of cells within annotated cell types defined by gene expression. SpliZ values for *SAT1* in blood classical monocytes had the largest Bhattacharyya distance among identified subpopulations (Materials and methods).

Junctional reads (defined by SpliZsites) driving the separation of the two subpopulations show distinct isoform expression profiles (*Figure 7C*): cells in cluster 1 splice to a 5' splice site that includes an ultraconserved genomic sequence, whereas those in cluster 2 contain splice to a different 5' splice site (*Figure 7C*). These clusters are not driven by *SAT1* gene expression and are not detected by gene-based clustering of monocytes as shown by visualization in cellxgene (*Megill et al., 2021*; *Figure 7C*). We used predictions of subpopulations of cells' splicing profiles in *SAT1* in individual 2 to blindly test whether classical monocytes in individual 1 also exhibited evidence of subpopulations based on their splicing profile. The number of cells with reads from the same junction – supporting subpopulations – is significantly greater than the number expected under the null assumption of randomly sampling two reads per cell regardless of cluster ($p$-value < 0.05, exact binomial test in individual 1, Materials and methods). Further supporting a biological role of *SAT1* splicing in the immune system, the same GMM-based approach identified two subpopulations of cells based on the SpliZ values for *SAT1* in both lung macrophages and thymus monocytes. Together with statistical support and blinded validation, this supports that *SAT1* exhibits splicing programs that define two splicing states within classical monocytes and calls for further investigations for up- and downstream regulation. Other genes including *PTP4A2*, *RABAC1*, and *MAGOH* have similar evidence of subpopulation structure and warrant further investigation.

## The SpliZ discovers conserved alternative splicing in mammalian spermatogenesis

The SpliZ provides a systematic framework to discover how splicing is regulated at a single-cell level along developmental trajectories (i.e., pseudotime). Previous studies have shown that testis is among the tissues with the highest isoform complexity and that even the isoform diversity in spermatogenic cells (spermatogonia, spermatocytes, round spermatids, and spermatozoa) is higher than that of many tissues (*Soumillon et al., 2013*). Also, RNA processing has been known to be important in spermatogenesis (*Green et al., 2018*). However, alternative splicing in single cells during spermatogenesis at the resolution of developmental time enabled by single-cell trajectory inference has not been studied. To systematically identify cells whose splicing is regulated during spermatogenesis, we applied the SpliZ to 4490 human sperm cells (*Hermann et al., 2018*) and compared findings to mouse (*Hermann et al., 2018*) and mouse lemur (*Tabula Microcebus Consortium, 2021*) sperm cells to test for conservation of regulated splicing changes.

170 genes out of 1757 genes with computable SpliZ in >100 human cells have splicing patterns that are significantly correlated with the pseudotime (|Spearman's correlation| > 0.1, Bonferroni-corrected $p$-value < 0.05, *Supplementary file 5*). The highest correlated genes included *TPPP2*, a

gene regulating tubulin polymerization implicated in male infertility (*Zhu et al., 2019*), *FAM71E1*, being predominantly expressed in testes (*Kwon et al., 2017*), *SPATA42*, a long non-coding RNA implicated in azoospermia (*Bo et al., 2020*), *MTFR1*, a gene regulating mitochondrial fission, and *MLF1*, an oncogene regulated in *Drosophila* testes (*Singh et al., 2016*; *Figure 8A*, *Figure 8—figure supplement 1*). In *MTFR1*, SpliZsites identify an unannotated 3' splice site in immature sperm showing evidence of novel transcripts (*Figure 8A*).

Among significantly correlated genes in human cells, splicing in 10 of these genes is also developmentally regulated in mouse and mouse lemur (*Supplementary file 5*). Centrosomal protein 112 (*CEP112*), a coiled-coil domain containing centrosomal protein and member of the cell division control protein, had the highest SpliZ-pseudotime correlation. It is highly expressed in testes and is essential for maintaining sperm function: loss-of-function mutations in *CEP112* have recently been associated with male infertility (*Sha et al., 2020*). Strikingly, the same 3' splice site and 5' splice sites identified by SpliZsites are orthologous and affect the apolipoprotein domain, a protein involved in the delivery of lipid between cell membranes and which is critical for the sperm development and fertility (*Setarehbadi et al., 2012*; *Figure 8B*).

*SPTY2D1OS* is another gene with conserved regulated splicing in spermatogenesis (*Figure 8—figure supplement 2*). Though highly expressed in human testes, it has unknown function in sperm development. *SPTY2D1OS* is located between *Uveld* and *SPTY2D* in the human genome; in mouse, *SPTY2D1OS* corresponds to a lncRNA named *Sirena1*, which has been recently shown to have function in mouse oocyte development (*Ganesh et al., 2020*) but has not previously been implicated in spermatogenesis. Together, our results suggest transcriptome-wide regulation of splicing in spermatogenic cells and call for more investigation into the function of splicing regulation not only in sperm development but also in other developmental trajectories.

## Discussion

Cell-type-specific splicing has been known to have functional effects in some cell types and in some genes for decades. However, technological limitations in measurement technology and methods to analyze resulting data have prevented high-throughput studies that profile the extent to which cell type can be predicted from splicing information alone. Full-coverage technologies such as SS2 have been the primary technologies for analyzing splicing in single cells thus far. However, SS2 is very difficult to scale: sequencing of 5000 cells that would take 2–3 days using 10X is estimated to take ~26 weeks using SS2 (*See et al., 2019*; *Svensson et al., 2020*). The lack of analysis of splicing in droplet data prevents discovery of regulated splicing in cell types that cannot be adequately profiled by plate-based approaches, and therefore causes biologically regulated splicing in these cell types to be missed (*Travaglini et al., 2020*). Here, we apply new analytic methodologies to find highly regulated splicing patterns from ubiquitous droplet-based sequencing platforms. These results reveal deeply conserved splicing programs that define tissue compartment and cell type in vivo.

Although the SpliZ method enables biological discovery of splicing differences based on droplet-based sequencing data, droplet-based data still presents major challenges for splicing analysis compared to full-length data. In this study, droplet-based sequencing has much lower sequencing coverage than full-length data, resulting in only 1416 genes with computable SpliZ values in the first human individual based on 10X data compared to 9802 genes with computable SpliZ values in SS2 data. Additionally, current droplet-based data is 3'-biased, meaning that some splicing events will never be sequenced by the technology and therefore cannot be analyzed. Despite these challenges, the ubiquity of droplet-based data, its utility for profiling rare cell types, and its unprecedented scale make it a powerful approach to discover regulated splicing.

The reproducibility of models in independently generated datasets suggests that the SpliZ can be applied globally to larger numbers of cell types to further identify splicing regulation at a single-cell level. We predict that as the number of cells profiled and the cell types grow, and global analyses are performed on data that is not 3' biased, the fraction of genes with evidence of cell-type-specific splicing will increase substantially beyond our current estimate of around 10% (*Figure 1F*). The results presented here lay the foundation for comprehensive splicing analysis in any scRNA-seq dataset and a reference to which future studies can be compared. Together, this work provides strong evidence for the hypothesis that alternative splicing in a large fraction of human genes is cell-type-specifically regulated and supports the idea that splicing is central to functional specialization of cell types.

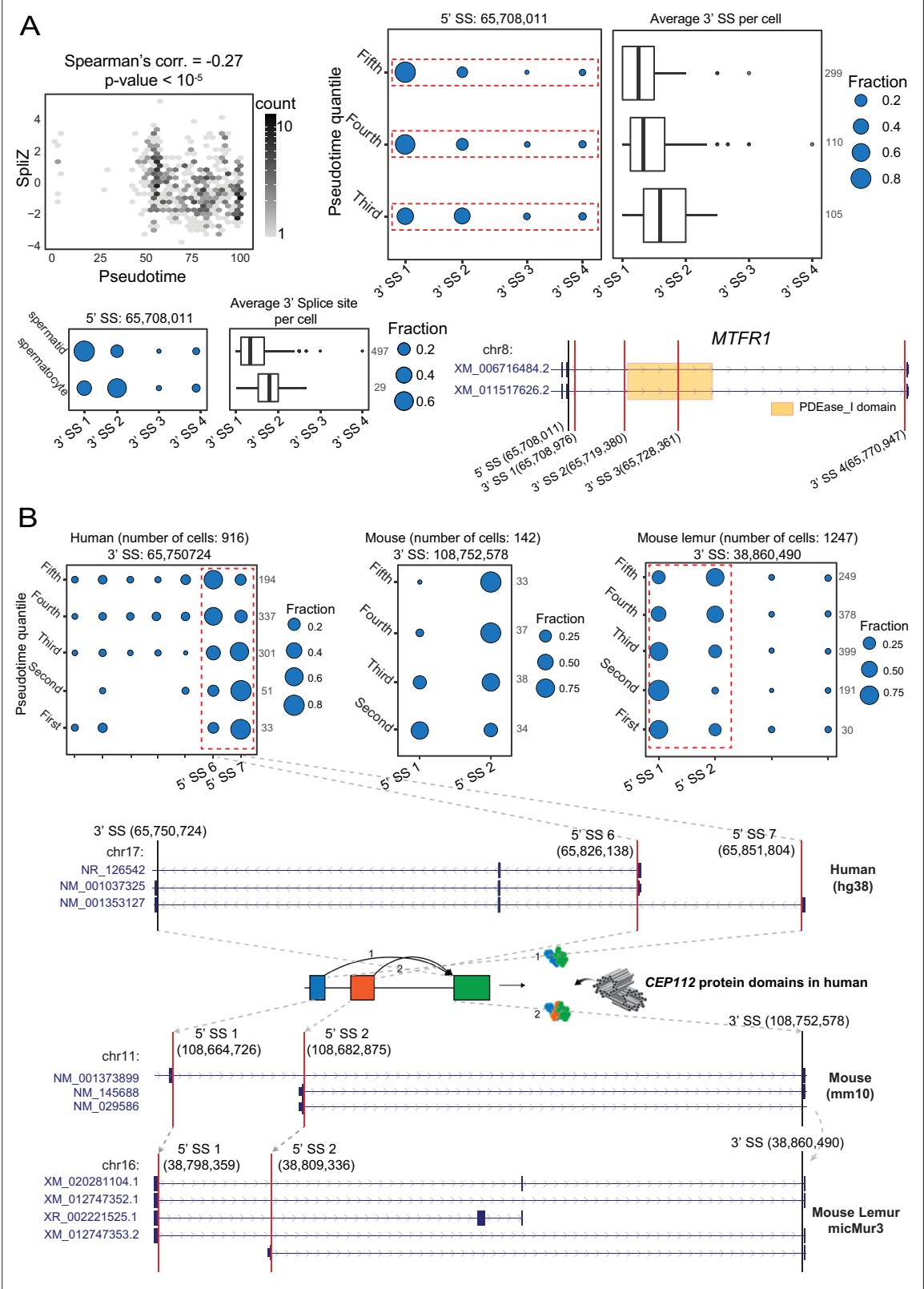

**Figure 8.** Developmentally regulated alternative splicing in mammalian spermatogenesis. (**A**) Regulated alternative splicing of *MTFR1* during sperm development. Significant negative correlation (Spearman's correlation = –0.27, p-value = 1.23e-7) between the SpliZ score for gene *MTFR1* and pseudotime in human sperm cells (top left). Dot plot and box plot show increasing use of a downstream 3′ splice site driving the *MTFR1* SpliZ in equal pseudotime quantiles (top right) with the same trend in immature (spermatocyte) and mature (spermatid) cells (bottom left). The gene structure for

*Figure 8 continued on next page*

*Figure 8 continued*

*MTFR1* according to the human RefSeq annotation database is shown in the bottom-right panel. The orange box on the gene structure represents the PDEase_I domain and how it is affected by the alternative splicing. (**B**) Dot plots showing the developmentally regulated alternative splicing of gene *CEP112* in testis cells from human, mouse, and mouse lemur. Cells are grouped according to pseudotime quantiles. The alternative splicing is conserved (i.e., involves the same set of 5' and 3' splice sites in human, mouse, and mouse lemur data as shown by the gray arrows on the gene structures) and involves 5' splice sites 65,826,138 and 65,851,804 in human, 5' splice sites 108,664,726 and 108,682,875 in mouse, and 5' splice sites 38,798,359 and 38,809,336 in mouse lemur. The 3' splice site and the two 5' splice sites involved in alternative splicing are shown by black and red vertical lines, respectively, on the gene structures. *CEP112* is on the minus strand in the human genome but is on the plus strand in mouse and mouse lemur genomes, leading to a negative correlation in splice site usage. Gray arrows show the LiftOver mapping between the 3' splice site and two 5' splice sites of the exon skipping event (indicating that the alternative splicing is conserved) and gray dashed lines for the human plot show the location of the 5' splice sites and how splicing changes the apolipoprotein protein domain.

The online version of this article includes the following figure supplement(s) for figure 8:

**Figure supplement 1.** Regulated alternative splicing of *MLF1* during sperm development in human, mouse, and mouse lemur.

**Figure supplement 2.** Regulated alternative splicing of *SPTY2D1OS* during sperm development in human and mouse lemur.

# Materials and methods

**Key resources table**

| Reagent type (species) or resource | Designation | Source or reference | Identifiers | Additional information |
|---|---|---|---|---|
| Software, algorithm | SICILIAN | *Dehghannasiri et al., 2021* | | https://github.com/salzmanlab/SICILIAN, *Roozbeh, 2021* |
| Software, algorithm | SpliZ Pipeline | *Olivieri et al., 2021* | | https://github.com/juliaolivieri/SpliZ_pipeline, *Julia, 2021b* |
| Software, algorithm | STAR | *Dobin et al., 2013* | | https://github.com/alexdobin/STAR, *Alexander, 2021* |
| Commercial assay or kit | BaseScope Duplex Reagent Kit - Hs | ACD (Bio-Techne) | cat. no 323,870 | |
| Commercial assay or kit | BaseScope Probes for *MYL6* | ACD (Bio-Techne) | BA-Hs-MYL6-tv1-1zz-st-C2 and BA-Hs-MYL6-tv2-1zz-st | |
| Commercial assay or kit | BaseScope Probes for *RPS24* | ACD (Bio-Techne) | BA-Hs-RPS24-tva-1zz-st-C2 and BA-Hs-RPS24-tvc-1zz-st | |
| Sequence-based reagent | FL-RPS24ex4F1 | This paper | PCR primer | /6FAM/CAATGTTGGTGCTGGCAAAA |
| Sequence-based reagent | RPS24ex6R2 | This paper | PCR primer | GCAGCACCTTTACTCCTTCGG |

## File downloads

- Human RefSeq hg38 annotation file was downloaded from ftp://ftp.ncbi.nlm.nih.gov/refseq/H_sapiens/annotation/GRCh38_latest/refseq_identifiers/GRCh38_latest_genomic.gff.gz
- Mouse lemur RefSeq Micmur3 annotation file was downloaded from https://www.ncbi.nlm.nih.gov/assembly/GCF_000165445.2/
- Mouse RefSeq GRCm38.p6 annotation file was downloaded from https://ftp.ncbi.nlm.nih.gov/genomes/all/GCF/000/001/635/GCF_000001635.26_GRCm38.p6/GCF_000001635.26_GRCm38.p6_genomic.gtf.gz
- The UCSC Pfam database for the human hg38 genome assembly was downloaded from http://hgdownload.soe.ucsc.edu/goldenPath/hg38/database/ucscGenePfam.txt.gz
- The Gene name mapping file for orthologous genes between human, mouse, and mouse lemur was downloaded from the Ensembl BioMart search tool (http://www.ensembl.org/biomart/martview/) on 12/11/2020.

## Code availability

Code to reproduce analysis and create figures is available through this GitHub repository: https://github.com/juliaolivieri/DiffSplice (copy archived at swh:1:rev:6fa54f473eb55c9e68692a6aa1d92d479e56b830, *Julia, 2021a*).

## Explanation of the SpliZ method

The SpliZ is a scalar score assigned to each cell-gene pair in a single-cell dataset. It is calculated using a three-step procedure (*Olivieri et al., 2021*). First, for every splice site with multiple partners in a dataset, those partners are assigned ranks according to their distance from the splice site. Next, each of these ranks is converted to a mean-zero, variance-one residual that quantifies the statistical deviation of that rank compared to the overall population. Finally, for a given cell and gene, these residuals are summed for each spliced read mapping to the corresponding splice site, and then scaled. Intuitively, the SpliZ for a particular gene has a large negative value if the introns for the gene in a given cell are smaller than average, and has a large positive value if the introns for the gene in a given cell are larger than average.

The SpliZVD is a modification of the SpliZ, in which rather than simply summing the splicing residuals for a given cell the residuals are scaled based on the eigenvector loadings of the first eigenvector of the residual matrix. The SVD decomposition of the residual matrix is also used to determine the SpliZsites for a given gene, which are defined as the three largest-magnitude components of the first eigenvector.

After SpliZ values are computed, if annotations are provided the SpliZ pipeline calculates which genes are differentially spliced between groups in the annotation. To calculate a p-value for whether the median SpliZ values by annotation are different for a given gene, the distribution of medians is first referred to as a null distribution. For p-values passing a nominal 0.05 level, permutations are performed to estimate the p-value with higher precision, and then adjusted using the Benjamini–Hochberg correction (*Hochberg and Benjamini, 1990*; *Olivieri et al., 2021*).

## SpliZ pipeline

Data from each individual was preprocessed from fastqs using SICILIAN with default parameters (*Dehghannasiri et al., 2021*). SICILIAN is a statistical method that can be applied to the BAM files by spliced aligners such as STAR (*Dobin et al., 2013*) to remove false-positive junction calls, enabling unbiased discovery of unannotated junctions that can contribute to alternative splicing. The scRNA-seq datasets were mapped using STAR version 2.7.5 a in two-pass mode with default parameters. Also, SICILIAN performs UMI deduplication to remove PCR duplicates. SpliZ scores were calculated using the SpliZ pipeline with default parameters (*Olivieri et al., 2021*). A SpliZ score was assigned to a gene-cell pair if there were at least five spliced reads from that gene aligned in that cell. Differential analysis was performed both based on tissue compartment (endothelial, epithelial, immune, and stromal) and independently based on cell type (defined by the tissue, compartment, and individual cell type, e.g., 'lung immune macrophage'). For a given gene, only cell types with at least 10 cells with computable SpliZ values for that gene were used. The SpliZ was used to call genes as significant for all datasets except the full SS2 datasets, for which the SpliZVD was used because of the increased complexity of full-length transcript data. We used a p-value cutoff of 0.05 after Benjamini–Hochberg correction. We define 'effect size' for a gene to be the largest magnitude median SpliZ (or SpliZVD) value out of all cell types with computable SpliZ for the gene. For between-cell-type-analysis, we use an effect size threshold of 0.5 (3.5 for SpliZVD) (Supplement) and require a difference of at least 0.5 within a single tissue and compartment for the gene to be called.

## FISH methods

Human rectus abdominis muscle biopsies from two donors were processed to single-cell suspensions by a combination of manual and enzymatical dissociation (*Tabula Sapiens Consortium, 2021*). Single-cell suspensions were stained with a combination of antibodies against CD45, CD31, THY1, and CD82, allowing for the isolation of immune cells (CD45+), endothelial cells (CD31+), mesenchymal cells (THY1+), and skeletal muscle satellite stem cells (CD82+). Due to the low number of immune cells present in the tissues, only the latter three cell types were stained. Cells were cytospun onto ECM-coated 8-well chamber slides and fixed in 4% PFA. Cells were washed in PBS and

prepared for RNA FISH by replacing the PBS to 100% ethanol. Cells were stained with custom probes according to the manufacturer's protocol (BaseScope Duplex Detection Reagent Kit [Advanced Cell Diagnostics, ACD]). Briefly, cells were rehydrated and treated with Protease IV solution (1;15 dilution) and were subsequently stained with indicated BaseScope probes for 2 hr in a hybridization oven set to 40 °C. Cells were then treated with amplification steps and imaged immediately after completion of the staining. As a control, human primary myoblasts were stained with the BaseScope probes and a negative control probe. Images were captured with a Zeiss Axiofluor microscope with collected CCD camera and a 40× objective lens. The red dye fluoresces in the 555 channel, whereas the green dye shows as gray in the DIC channel. Images were quantified with Volocity software. One muscle sample was independently fixed in 10% neutral buffered formalin for 24 hr in preparation for BaseScope staining in cryosections. Tissue was dehydrated in 20% sucrose for 24 hr, washed in PBS, dried, embedded in OCT, and frozen in cooled isopentane. Sections of 10 μm were cut and dried in –20 °C for 1 hr and stored –80 °C until use. Tissue slides were removed from –80 °C and immediately washed with PBS to remove OCT, dried, and baked in 60 °C for 30 min. Tissue slides were post-fixed in 4% PFA for 15 min and dehydrated by immersing slides in 50, 70, and 100% ethanol for 5 min each. Tissue slides were then treated with hydrogen peroxide for 10 min and washed briefly with distilled water and subjected to target retrieval for 5 min, washed briefly with distilled water and in 100% ethanol. Tissue slides were treated with Protease IV solution in 40 °C for 10 min and washed twice with distilled water and hybridized with indicated BaseScope probes for 2 hr in 40 °C. Tissue slides were then treated with amplification steps. For dual FISH and IHC staining, tissue slides were immediately blocked in blocking buffer for 30 min (5% FBS, 1% BSA, 0.1% Triton-X100, 0.01% sodium azide in PBS) and stained with Pax7 antibody (1:100) in blocking buffer overnight in 4 °C. Tissue slides were washed with 0.1% Tween-20 in PBS three times and then fluorescently conjugated secondary antibodies were added for an hour in room temperature. After three additional washes, tissue slides were dried, mounted, and imaged immediately.

Deidentified human adult lung tissue was obtained from the Stanford Tissue Bank. The tissue was fixed in 10% neutral buffered formalin and embedded in paraffin. For the single-molecule in situ hybridization, 6-μm-thick paraffin sections were prepared and processed following the BaseScope Duplex Detection Reagent Kit (ACD) protocol, modified to use brown DAB chromogen in place of the usual green chromogen as the second color (custom protocol from ACD). Stained slides were visualized using an Olympus upright bright field microscope at 20× and 40× magnification. Cell types were identified by a pathologist based on cell morphology highlighted by the hematoxylin counterstain. Representative images of each cell type of interest were captured using an Olympus digital microscope color camera. Quantification was done by demarcating a polygonal image region containing multiple cells of homogeneous type and manually counting all the dots of each color within the region.

Proprietary probes (ACD) used for both human lung and muscle: BA-Hs-RPS24-tvc-1zz-st (targets 400–437 of NM_001026.5), BA-Hs-RPS24-tva-1zz-st (targets 399–437 of NM_033022.4); BA-Hs-MYL6-tv1-1zz-st (targets 469–505 of NM_021019.5), BA-Hs-MYL6-tv2-1zz-st (targets 436–480 of NM_079423.4).

## Single-cell RT-PCR

SS2 preamplified cDNA of single cells from the Human Lung Cell Atlas project (*Travaglini et al., 2020*) was used as starting templates. The cells correspond to wells N14, A16, H14, B6, A3, A7, A13, A11, A8, D1, A12, A17, B12, J16, A21, P22, D23, A22, B22 of plate B002014; cell type metadata was taken from https://www.synapse.org/#!Synapse:syn21041850/wiki/60086. 1 μl of primary preamp was further preamplified in a 20 μl reaction (100 nM ISPCR primer = AAGCAGTGGTATCAACGCAGAGT, KAPA HiFi Fidelity mix; program: 95° 3'; 9 × [98° 20"; 67° 15"; 72° 4']; 72° 5'), then diluted eightfold with water. 2 μl of this secondary preamp was used as template in a 40 μl reaction (500 nM each of primers FL-RPS24ex4F1 = /6FAM/CAATGTTGGTGCTGGCAAAA and RPS24ex6R2 = GCAGCACC TTTACTCCTTCGG, New England Biolabs Phusion HF buffer, 200 nM dNTPs, 0.4 units Phusion DNA Polymerase; program: 98° 30"; 24 × [98° 10"; 60° 15"; 72° 20"]; 72° 5'). PCRs were diluted 1:100 and run on an ABI 3130xl Genetic Analyzer with GS500ROX standard; peaks were called by the Thermo Fisher Cloud Peak Scanner app, and presented as a pseudo-gel image using a custom Python script. For Sanger sequencing, secondary preamps were used in a similar PCR but with primers RPS24ex4F4 = AAGCAACGAAAGGAACGCAA and RPS24ex6R4 = CCACAGCTAACATCATTGCAG; the cleaned-up

products were sequenced with the same primers. Oligonucleotide synthesis and capillary electrophoresis were done by Stanford PAN (Protein and Nucleic Acid Facility).

## Concordance analysis between technologies and donors

Concordance with SS2 was used as an extra test of the reproducibility of the method. SS2 and 10X datasets were subset to include only junctions and cell types shared in both to make the datasets as comparable as possible, and remove RNA measurements that could only be detected by SS2. Next, the SpliZ was calculated independently for both datasets as described for 10X . We then correlated the median SpliZ scores for matched genes and ontologies for genes called as significant by both technologies in the same individual. This resulted in a Pearson correlation of 0.439 between the two technologies for individual 1 and 0.769 for individual 2. We similarly subsetted both 10X datasets so that they each only included shared cell types and junctions, and then ran the SpliZ pipeline separately on each dataset, resulting in a Pearson correlation of 0.776 between the two datasets.

## K-means clustering of *RPS24* and *ATP5F1C*

We first subsetted to only cells in the immune, epithelial, and stromal compartments with computable SpliZ values for both *RPS24* and *ATP5F1C* in the 10X data (9712 cells in individual 1, 2370 cells in individual 2). K-means clustering was performed with the "sklearn" package in Python with k = 3 to separate all cells into three clusters based on their *RPS24* and *ATP5F1C* SpliZ values. Each resulting cluster was assigned to one compartment such as to minimize classification error, and accuracy was calculated for each compartment based on these cluster assignments.

## LiftOver shared sites

We used the UCSC LiftOver tool (https://genome.ucsc.edu/cgi-bin/hgLiftOver) with the recommended settings to convert the coordinates between human (hg38), mouse (mm10), and mouse lemur (Mmur3). To find shared SpliZsites between human, mouse, and mouse lemur, we subset to only those junctions that had been successfully and uniquely converted by the LiftOver tool.

## Spermatogenesis analysis

To find genes with regulated splicing during sperm development, for each gene, Spearman's correlation was computed between the SpliZ and pseudotime values across the cells with computable SpliZ scores for that gene. We considered only genes with computable SpliZ in at least 100 cells, and for each organism (human, mouse, mouse lemur), the genes with |Spearman's coefficient| >0.1 and Bonferroni-corrected p-value < 0.05 were selected as significantly splicing regulated genes. Only those genes that have names in all three organisms were considered for the conservation analysis.

## Subpopulation analysis

To find subcluster of cells within cell types that can be distinguished based on the splice profile as quantified by the SpliZ score, we take advantage of the fact that under the null hypothesis all cells within a cell type should follow a univariate normal distribution for the SpliZ score of each gene; however, if there are subcluster of cells with distinct splice profiles, the distribution should be better modeled via a GMM. To find the optimal number of components for the distribution of the SpliZ score for each gene within a cell type, we used the ICL, which is a model selection criterion, and selected the optimal number of component as the number that attains the knee point in the ICL curve for different component numbers. If ICL selects at least two subclusters within a cell type, we assigned cells to one of the clusters based on the fitted GMM with the optimal number of components. After clustering cells, we checked to see if subclusters are disjoint enough by computing the Bhattacharyya distance between the subclusters. The subclusters for a pair of gene and cell type are called, if the distance between subclusters is >0.5. Otherwise, we reduced the optimal number of components by 1 and then again ran the GMM clustering to see if the new subclusters can satisfy the Bhattacharyya criterion. We keep doing this until either we end up with only one cluster (which means no subcluster is found) or the resulting subclusters have enough distance.

To test whether the subpopulations of *SAT1* were the result of a 'bimodal splicing' artifact as reported in *Buen Abad Najar et al., 2020*, we performed the following analysis. Considering the blood classical monocytes in human individual 1 together, we calculated the fraction $p$ of junctional

reads aligning to the 5′ splice site 23,785,328 in *SAT1* that partner with the 3′ splice site 23,783,883 rather than 23,784,403. We found p = 83/102 = 0.814 (the probability was 0.893 in individual 2). We then subset to only cells with exactly two reads mapping to 5′ splice site 23,785,328, resulting in 16 cells. Out of these 16 cells, all had either both reads mapping to 3′ splice site 23,783,883 or both reads mapping to 3′ splice site 23,784,403. We calculated the probability of zero cells having one read mapping to each splice site under the null hypothesis as follows: (1 - binom.pmf(1,2,0.814))[16] = 0.00312 (exact binomial test). Because processing of 10X data includes a UMI deduplication step through SICILIAN (*Dehghannasiri et al., 2021*), these duplicates are not PCR duplicates.

## SpliZsite analysis

23 out of 148 (fraction = 0.1554) and 11 out of 138 (fraction = 0.0797) SpliZsites found in human individual 1 were also found to be SpliZsites in mouse lemur and mouse, respectively. We limited the comparison with mouse (resp. mouse lemur) SpliZsites to only those SpliZsites whose corresponding genes had computable SpliZ in mouse (resp. mouse lemur). To compute the expected fraction of shared SpliZsites between human and one of the other organisms under the null, we first need the null probability of each SpliZsite being shared between human and the other organism by considering the number of splice sites for the gene. This probability is $\frac{1}{N_i}$, where $N_i$ is the number of distinct splice sites (we considered only 5′ splice sites) with junctional reads according to SICILIAN. If there are $I$ SpliZsites in human whose genes have also computable SpliZ scores in the other organism, the expected fraction of shared SpliZsites between human and that organism is $\frac{1}{I} \sum_{i=1}^{I} \frac{1}{N_i}$, resulting in the expected fractions 0.071 and 0.088 of human SpliZsites shared with mouse lemur and mouse, respectively. The p-value for the observed fraction 0.1554 for mouse lemur can be approximated using a binomial test (the binomial test is an approximation as the success probability for each SpliZsite changes according to $\frac{1}{N_i}$), which results in a p-value of 0.0003.

## Choosing effect size filters

To choose the filters for differential analysis in 10X data, we subsetted the TSP1 and TSP2 10X data to only junctions shared in both individuals and cell types shared in both individuals. We then ran the SpliZ pipeline. Using a p-value threshold of 0.05 for the SpliZ, we tested the correlation between median SpliZ values matching on cell type for genes called as significant in both datasets based on that effect size cutoff. Without an effect size cutoff, there was already a correlation of 0.2. We chose the effect size threshold 0.5 because it yielded a correlation of 0.6 between the datasets (*Figure 8— figure supplement 1*). To decide the SpliZVD cutoff for SS2 data, we perform the same procedure, except this time use the SS2 data from TSP1 and TSP2, again restricted to only shared junctions and shared cell types. Because the correlation never reaches 0.6 for this data, we choose a cutoff of 3.5 because it maximizes correlation (*Figure 8—figure supplement 1*).

## Sanger sequencing confirms low levels of the +*a*-*b*+*c RPS24* isoform in mouse kidney

Whole-tissue RNA was amplified from mouse adult kidney and human fetal kidney, assuming that a large fraction of the RNA would be coming from epithelial cells. The PCR products were cloned and then ligated so that multiple could be read out per each Sanger read. However due to the repetitive nature of the inserts, the read quality was poor and only the first couple of inserts could be interpreted. Generally, our results confirm that +a-b+c is not very abundant in mouse: in human kidney, it is 45% of total, while in mouse kidney it is only 14% (data not shown).

## Bowtie2 alignment of *RPS24* reads

Custom fasta files were created for human, mouse, and mouse lemur. Each includes eight transcripts corresponding to all combinations of inclusion of the a , b, and c exons in *RPS24*. Each sequence is centered on whichever of these exons are included, and padded on either side with sequence from exon 4 and exon 6 such that each sequence is 150 base pairs long. A Bowtie2 index was created based on these transcriptomes for each species, and all fasta files for individual 1 from human, mouse, and mouse lemur were aligned to the respective index using Bowtie2 with the command `bowtie2 --no-unal index/{params.species}_RPS24_bwt -U {input} -S {output}`, where {params.

`species}` is the name of the species' index, `{input}` is the input fasta and `{output}` is the output file name. Because exons a , b, and c are 3, 18, and 22 base pairs long, respectively, and 10X reads are around 90 base pairs long, each read aligning to one of these transcripts uniquely identifies the isoform.

## Enrichment of genes significant in all three species analysis

We test whether the number of genes significant in all three species is more than expected under the null hypothesis, which is that there is no evolutionary conservation between the species. Let $s$ be the number of genes that are shared between human, lemur, and mouse that are significant at least once in all three species. Let $n$ be the total number of genes present in at least 20 cells in a cell type in all three species (note: mapping between species is not perfect, so some genes present in all are probably missing). The probability that a given gene is significant in all three species is

$$P(gene\ sig\ in\ 3) = P(gene\ sig\ in\ human)P(gene\ sig\ in\ lemur\ |\ gene\ sig\ in\ human)$$

$$P(gene\ sig\ in\ mouse\ |\ gene\ sig\ in\ human,\ gene\ sig\ in\ lemur).$$

Under the null hypothesis, assume that a gene being significant in one species is independent from it being significant in either other species. Therefore, under the null hypothesis, $P(gene\ sig\ in\ 3) = P(gene\ sig\ in\ human)P(gene\ sig\ in\ lemur)P(gene\ sig\ in\ mouse)$.

We can estimate the quantities on the right-hand side of the question for each species. For every species, calculate $p_{species}$, where

$$p_{species} = \frac{(\#\ genes\ that\ ever\ have\ a\ significant\ mz\ in\ species)}{(total\ \#\ genes\ with\ computable\ SpliZ\ in\ at\ least\ 20\ cells\ in\ a\ cell\ type)}.$$

Therefore, under the null hypothesis, the probability that at least $x$ genes are significant in all three species out of $n$ genes is given by $1 - binom\_cdf(x, n, p_{human} * p_{lemur} * p_{mouse})$. The estimates of the degree of regulated splicing are a lower bound as they are based on (a) sampling only a subset of organs and (b) based on studying only a subset of genes that are sampled sufficiently with current sequencing depth conventions. Incomplete gene naming conventions, especially in the mouse and lemur, may restrict the power of this analysis.

## Acknowledgements

We thank Manny Ares, Douglas Black, Maria Barna, and members of the Salzman lab for insightful discussions. We thank Kyle Travaglini (Krasnow lab) for aliquots of single-cell RT-PCR preamplification from the Human Lung Cell Atlas samples. We thank Jessica Klein for creating part of *Figure 1*. JO is supported by the National Science Foundation Graduate Research Fellowship under Grant No. DGE-1656518 and a Stanford Graduate Fellowship. RD is supported by the Cancer Systems Biology Scholars Program Grant R25 CA180993 and the Clinical Data Science Fellowship Grant T15 LM7033-36. JS is supported by the National Institute of General Medical Sciences Grant R01 GM116847 and NSF Faculty Early Career Development Program Award MCB1552196. None of these funding sources were involved in study design, data collection and interpretation, or the decision to submit the work for publication.

## Additional information

### Group author details

**Tabula Sapiens Consortium**

**Robert C Jones**: Department of Bioengineering, Stanford University, Stanford, United States; **Jim Karkanias**: Chan Zuckerberg Biohub, San Francisco, United States; **Mark Krasnow**: Department of Biochemistry, Stanford University School of Medicine, Stanford, United States; Howard Hughes Medical Institute, Chevy Chase, United States; **Angela Oliveira Pisco**: Chan Zuckerberg Biohub, San Francisco, United States; **Stephen Quake**: Department of Bioengineering, Stanford University, Stanford, United States; Chan Zuckerberg Biohub, San Francisco, United States; Department of Applied

Physics, Stanford University, Stanford, United States; **Julia Salzman**: Howard Hughes Medical Institute, Chevy Chase, United States; Department of Biomedical Data Science, Stanford University, Stanford, United States; **Nir Yosef**: Chan Zuckerberg Biohub, San Francisco, United States; Center for Computational Biology, University of California Berkeley, Berkeley, United States; Department of Electrical Engineering and Computer Sciences, University of California Berkeley, Berkeley, United States; Ragon Institute of MGH, MIT and Harvard, Cambridge, United States; **Bryan Bulthaup**: Donor Network West, San Ramon, United States; **Phillip Brown**: Donor Network West, San Ramon, United States; **Will Harper**: Donor Network West, San Ramon, United States; **Marisa Hemenez**: Donor Network West, San Ramon, United States; **Ravikumar Ponnusamy**: Donor Network West, San Ramon, United States; **Ahmad Salehi**: Donor Network West, San Ramon, United States; **Bhavani Sanagavarapu**: Donor Network West, San Ramon, United States; **Eileen Spallino**: Donor Network West, San Ramon, United States; **Ksenia A Aaron**: Department of Otolaryngology-Head and Neck Surgery, Stanford University School of Medicine, Stanford, United States; **Waldo Concepcion**: Donor Network West, San Ramon, United States; **James M Gardner**: Department of Surgery, University of California San Francisco, San Francisco, United States; Diabetes Center, University of California San Francisco, San Francisco, United States; **Burnett Kelly**: Donor Network West, San Ramon, United States; DCI Donor Services, Sacramento, United States; **Nikole Neidlinger**: Donor Network West, San Ramon, United States; **Zifa Wang**: Donor Network West, San Ramon, United States; **Sheela Crasta**: Department of Bioengineering, Stanford University, Stanford, United States; Chan Zuckerberg Biohub, San Francisco, United States; **Saroja Kolluru**: Department of Bioengineering, Stanford University, Stanford, United States; Chan Zuckerberg Biohub, San Francisco, United States; **Maurizio Morri**: Chan Zuckerberg Biohub, San Francisco, United States; **Serena Y Tan**: Department of Pathology, Stanford University School of Medicine, Stanford, United States; **Kyle J Travaglini**: Department of Biochemistry, Stanford University School of Medicine, Stanford, United States; **Chenling Xu**: Center for Computational Biology, University of California Berkeley, Berkeley, United States; **Marcela Alcántara-Hernández**: Department of Microbiology and Immunology, Stanford University, Stanford, United States; **Nicole Almanzar**: Department of Pediatrics - Pulmonary Medicine, Stanford University, Stanford, United States; **Jane Antony**: Institute for Stem Cell Biology and Regenerative Medicine, Stanford University School of Medicine, Stanford, United States; **Benjamin Beyersdorf**: Department of Medicine, Division of Cardiovascular Medicine, Stanford University, Stanford, United States; **Deviana Burhan**: Department of Medicine and Liver Center, University of California San Francisco, San Francisco, United States; **Kruti Calcuttawala**: Institute for Stem Cell Biology and Regenerative Medicine, Stanford University School of Medicine, Stanford, United States; **Mathew Carter**: Department of Microbiology and Immunology, Stanford University, Stanford, United States; **Charles KF Chan**: Institute for Stem Cell Biology and Regenerative Medicine, Stanford University School of Medicine, Stanford, United States; Department of Surgery - Plastic and Reconstructive Surgery, Stanford University School of Medicine, Stanford, United States; **Charles A Chang**: Department of Developmental Biology, Stanford University School of Medicine, Stanford, United States; **Alex Colville**: Department of Neurology and Neurological Sciences, Stanford University School of Medicine, Stanford, United States; Paul F. Glenn Center for the Biology of Aging, Stanford University School of Medicine, Stanford, United States; **Rebecca Culver**: Department of Microbiology and Immunology, Stanford University, Stanford, United States; **Ivana Cvijović**: Department of Bioengineering, Stanford University, Stanford, United States; Department of Applied Physics, Stanford University, Stanford, United States; **Gaetano D'Amato**: Department of Biology, Stanford University, Stanford, United States; **Camille Ezran**: Department of Biochemistry, Stanford University School of Medicine, Stanford, United States; **Francisco Galdos**: Institute for Stem Cell Biology and Regenerative Medicine, Stanford University School of Medicine, Stanford, United States; **Astrid Gillich**: Department of Biochemistry, Stanford University School of Medicine, Stanford, United States; **William R Goodyer**: Department of Pediatrics, Division of Cardiology, Stanford University School of Medicine, Stanford, United States; **Yan Hang**: Department of Developmental Biology, Stanford University School of Medicine, Stanford, United States; **Alyssa Hayashi**: Department of Bioengineering, Stanford University, Stanford, United States; **Sahar Houshdaran**: Center for Gynecology and Reproductive Sciences, Department of Obstetrics, Gynecology and Reproductive Sciences, University of California San Francisco, San Francisco, United States; **Xianxi Huang**: Department of Medicine, Division of Cardiovascular Medicine, Stanford University, Stanford, United States; Department of Critical Care Medicine, The First Affiliated Hospital of Shantou University Medical College, Shantou,

China; **Juan Irwin**: Center for Gynecology and Reproductive Sciences, Department of Obstetrics, Gynecology and Reproductive Sciences, University of California San Francisco, San Francisco, United States; **SoRi Jang**: Department of Biochemistry, Stanford University School of Medicine, Stanford, United States; **Julia Vallve Juanico**: Center for Gynecology and Reproductive Sciences, Department of Obstetrics, Gynecology and Reproductive Sciences, University of California San Francisco, San Francisco, United States; **Aaron M Kershner**: Institute for Stem Cell Biology and Regenerative Medicine, Stanford University School of Medicine, Stanford, United States; **Soochi Kim**: Department of Neurology and Neurological Sciences, Stanford University School of Medicine, Stanford, United States; Paul F. Glenn Center for the Biology of Aging, Stanford University School of Medicine, Stanford, United States; **Bernhard Kiss**: Institute for Stem Cell Biology and Regenerative Medicine, Stanford University School of Medicine, Stanford, United States; **William Kong**: Institute for Stem Cell Biology and Regenerative Medicine, Stanford University School of Medicine, Stanford, United States; **Maya E Kumar**: Sean N. Parker Center for Asthma and Allergy Research, Stanford University School of Medicine, Stanford, United States; **Rebecca Leylek**: Department of Microbiology and Immunology, Stanford University, Stanford, United States; **Baoxiang Li**: Department of Ophthalmology, Stanford University School of Medicine, Stanford, United States; **Shixuan Liu**: Department of Biochemistry, Stanford University School of Medicine, Stanford, United States; **Gabriel Loeb**: Division of Nephrology, Department of Medicine, University of California San Francisco, San Francisco, United States; **Wan-Jin Lu**: Institute for Stem Cell Biology and Regenerative Medicine, Stanford University School of Medicine, Stanford, United States; **Jonathan Maltzman**: Division of Nephrology, Stanford University School of Medicine, Stanford, United States; Veterans Administration Palo Alto Health Care System and Department of Medicine, Palo Alto, United States; **Sruthi Mantri**: Stanford University School of Medicine, Stanford, United States; **Maxim Markovic**: Department of Bioengineering, Stanford University, Stanford, United States; **Patrick L McAlpine**: Mass Spectrometry Platform, Chan Zuckerberg Biohub, Stanford, United States; **Ross Metzger**: Department of Pediatrics, Division of Cardiology, Stanford University School of Medicine, Stanford, United States; Vera Moulton Wall Center for Pulmonary and Vascular Disease, Stanford University School of Medicine, Stanford, United States; **Antoine de Morree**: Department of Neurology and Neurological Sciences, Stanford University School of Medicine, Stanford, United States; Paul F. Glenn Center for the Biology of Aging, Stanford University School of Medicine, Stanford, United States; **Karim Mrouj**: Institute for Stem Cell Biology and Regenerative Medicine, Stanford University School of Medicine, Stanford, United States; **Shravani Mukherjee**: Department of Ophthalmology, Stanford University School of Medicine, Stanford, United States; **Tyler Muser**: Department of Pediatrics - Pulmonary Medicine, Stanford University, Stanford, United States; **Patrick Neuhöfer**: Stanford Cancer Institute, Stanford University School of Medicine, Stanford, United States; **Thi Nguyen**: Division of Nephrology, Department of Medicine, University of California San Francisco, San Francisco, United States; **Kimberly Perez**: Department of Microbiology and Immunology, Stanford University, Stanford, United States; **Ragini Phansalkar**: Department of Biology, Stanford University, Stanford, United States; **Nazan Puluca**: Institute for Stem Cell Biology and Regenerative Medicine, Stanford University School of Medicine, Stanford, United States; **Zhen Qi**: Institute for Stem Cell Biology and Regenerative Medicine, Stanford University School of Medicine, Stanford, United States; **Poorvi Rao**: Department of Medicine and Liver Center, University of California San Francisco, San Francisco, United States; **Hayley Raquer-McKay**: Department of Microbiology and Immunology, Stanford University, Stanford, United States; **Koki Sasagawa**: Department of Medicine, Division of Cardiovascular Medicine, Stanford University, Stanford, United States; **Nicholas Schaum**: Institute for Stem Cell Biology and Regenerative Medicine, Stanford University School of Medicine, Stanford, United States; Department of Neurology and Neurological Sciences, Stanford University School of Medicine, Stanford, United States; **Bronwyn Lane Scott**: Department of Ophthalmology, Stanford University School of Medicine, Stanford, United States; **Bobak Seddighzadeh**: Division of Hematology and Oncology, Department of Medicine, University of California San Francisco, San Francisco, United States; **Joe Segal**: Department of Medicine and Liver Center, University of California San Francisco, San Francisco, United States; **Sushmita Sen**: Center for Gynecology and Reproductive Sciences, Department of Obstetrics, Gynecology and Reproductive Sciences, University of California San Francisco, San Francisco, United States; **Sean Spencer**: Department of Medicine - Med/Gastroenterology and Hepatology, Stanford University School of Medicine, Stanford, United States; **Lea Steffes**: Department of Pediatrics - Pulmonary Medicine, Stanford University, Stanford, United States; **Varun R**

**Subramaniam**: Department of Ophthalmology, Stanford University School of Medicine, Stanford, United States; **Aditi Swarup**: Department of Ophthalmology, Stanford University School of Medicine, Stanford, United States; **Michael Swift**: Department of Bioengineering, Stanford University, Stanford, United States; **Will Van Treuren**: Department of Microbiology and Immunology, Stanford University, Stanford, United States; **Emily Trimm**: Department of Biology, Stanford University, Stanford, United States; **Maggie Tsui**: Department of Medicine and Liver Center, University of California San Francisco, San Francisco, United States; **Stefan Veizades**: Department of Medicine, Division of Cardiovascular Medicine, Stanford University, Stanford, United States; Stanford Cardiovascular Institute, Stanford, United States; College of Medicine and Veterinary Medicine, University of Edinburgh, Edinburgh, United Kingdom; **Sivakamasundari Vijayakumar**: Institute for Stem Cell Biology and Regenerative Medicine, Stanford University School of Medicine, Stanford, United States; **Kim Chi Vo**: Center for Gynecology and Reproductive Sciences, Department of Obstetrics, Gynecology and Reproductive Sciences, University of California San Francisco, San Francisco, United States; **Sevahn K Vorperian**: Department of Bioengineering, Stanford University, Stanford, United States; **Hannah Weinstein**: Division of Hematology and Oncology, Department of Medicine, University of California San Francisco, San Francisco, United States; **Juliane Winkler**: Department of Cell & Tissue Biology, University of California San Francisco, San Francisco, United States; **Timothy TH Wu**: Department of Biochemistry, Stanford University School of Medicine, Stanford, United States; **Jamie Xie**: Division of Hematology and Oncology, Department of Medicine, University of California San Francisco, San Francisco, United States; **Andrea R Yung**: Department of Biochemistry, Stanford University School of Medicine, Stanford, United States; **Yue Zhang**: Department of Biochemistry, Stanford University School of Medicine, Stanford, United States; **Angela M Detweiler**: Chan Zuckerberg Biohub, San Francisco, United States; **Honey Mekonen**: Chan Zuckerberg Biohub, San Francisco, United States; **Norma Neff**: Chan Zuckerberg Biohub, San Francisco, United States; **Rene V Sit**: Chan Zuckerberg Biohub, San Francisco, United States; **Michelle Tan**: Chan Zuckerberg Biohub, San Francisco, United States; **Jia Yan**: Chan Zuckerberg Biohub, San Francisco, United States; **Gregory R Bean**: Department of Pathology, Stanford University School of Medicine, Stanford, United States; **Gerald J Berry**: Department of Pathology, Stanford University School of Medicine, Stanford, United States; **Vivek Charu**: Department of Pathology, Stanford University School of Medicine, Stanford, United States; **Erna Forgó**: Department of Pathology, Stanford University School of Medicine, Stanford, United States; **Brock A Martin**: Department of Pathology, Stanford University School of Medicine, Stanford, United States; **Michael G Ozawa**: Department of Pathology, Stanford University School of Medicine, Stanford, United States; **Oscar Silva**: Department of Pathology, Stanford University School of Medicine, Stanford, United States; **Angus Toland**: Department of Pathology, Stanford University School of Medicine, Stanford, United States; **Venkata NP Vemuri**: Chan Zuckerberg Biohub, San Francisco, United States; **Shaked Afik**: Center for Computational Biology, University of California Berkeley, Berkeley, United States; **Rob Bierman**: Department of Biochemistry, Stanford University School of Medicine, Stanford, United States; **Olga Borisovna Botvinnik**: Chan Zuckerberg Biohub, San Francisco, United States; **Ashley Byrne**: Chan Zuckerberg Biohub, San Francisco, United States; **Michelle Chen**: Department of Bioengineering, Stanford University, Stanford, United States; **Roozbeh Dehghannasiri**: Department of Biochemistry, Stanford University School of Medicine, Stanford, United States; Department of Biomedical Data Science, Stanford University, Stanford, United States; **Adam Gayoso**: Center for Computational Biology, University of California Berkeley, Berkeley, United States; **Alejandro A Granados**: Chan Zuckerberg Biohub, San Francisco, United States; **Qiqing Li**: Chan Zuckerberg Biohub, San Francisco, United States; **Gita Mahmoudabadi**: Department of Bioengineering, Stanford University, Stanford, United States; **Aaron McGeever**: Chan Zuckerberg Biohub, San Francisco, United States; **Julia Eve Olivieri**: Department of Biochemistry, Stanford University School of Medicine, Stanford, United States; Department of Biomedical Data Science, Stanford University, Stanford, United States; Institute for Computational and Mathematical Engineering, Stanford University, Stanford, United States; **Madeline Park**: Chan Zuckerberg Biohub, San Francisco, United States; **Neha Ravikumar**: Department of Bioengineering, Stanford University, Stanford, United States; **Julia Salzman**: Department of Biomedical Data Science, Stanford University, Stanford, United States; **Sandra L Schmid**: Chan Zuckerberg Biohub, San Francisco, United States; **Geoff Stanley**: Department of Bioengineering, Stanford University, Stanford, United States; **Weilun Tan**: Chan Zuckerberg Biohub, San Francisco, United States; **Alexander J Tarashansky**: Chan Zuckerberg Biohub, San Francisco, United States; **Rohan Vanheusden**:

Chan Zuckerberg Biohub, San Francisco, United States; **Sheng Wang**: Chan Zuckerberg Biohub, San Francisco, United States; **Galen Xing**: Chan Zuckerberg Biohub, San Francisco, United States; **Nir Yosef**: Department of Biomedical Data Science, Stanford University, Stanford, United States; Department of Electrical Engineering and Computer Sciences, University of California Berkeley, Berkeley, United States; **Les Dethlefsen**: Division of Infectious Diseases & Geographic Medicine, Department of Medicine, Stanford University School of Medicine, Stanford, United States; **Po-Yi Ho**: Department of Microbiology and Immunology, Stanford University, Stanford, United States; **Juan C Irwin**: Center for Gynecology and Reproductive Sciences, Department of Obstetrics, Gynecology and Reproductive Sciences, University of California San Francisco, San Francisco, United States; **Maya E Kumar**: Department of Pediatrics - Pulmonary Medicine, Stanford University, Stanford, United States; **Angera H Kuo**: Institute for Stem Cell Biology and Regenerative Medicine, Stanford University School of Medicine, Stanford, United States; **Patrick Neuhöfer**: Stanford Cancer Institute, Stanford University School of Medicine, Stanford, United States; **Kimberly Perez**: Department of Microbiology and Immunology, Stanford University, Stanford, United States; **Hayley Raquer-McKay**: Department of Microbiology and Immunology, Stanford University, Stanford, United States; **Rahul Sinha**: Department of Pathology, Stanford University School of Medicine, Stanford, United States; Institute for Stem Cell Biology and Regenerative Medicine, Stanford University School of Medicine, Stanford, United States; Stanford Cancer Institute, Stanford University School of Medicine, Stanford, United States; **Hanbing Song**: Division of Hematology and Oncology, Department of Medicine, University of California San Francisco, San Francisco, United States; **Sean Spencer**: Department of Medicine - Med/Gastroenterology and Hepatology, Stanford University School of Medicine, Stanford, United States; **Bruce Wang**: Department of Medicine and Liver Center, University of California San Francisco, San Francisco, United States; **Juliane Winkler**: Department of Cell & Tissue Biology, University of California San Francisco, San Francisco, United States; **Steven E Artandi**: Department of Biochemistry, Stanford University School of Medicine, Stanford, United States; Stanford Cancer Institute, Stanford University School of Medicine, Stanford, United States; **Philip Beachy**: Department of Developmental Biology, Stanford University School of Medicine, Stanford, United States; **Michael F Clarke**: Institute for Stem Cell Biology and Regenerative Medicine, Stanford University School of Medicine, Stanford, United States; **Linda Giudice**: Center for Gynecology and Reproductive Sciences, Department of Obstetrics, Gynecology and Reproductive Sciences, University of California San Francisco, San Francisco, United States; **Franklin Huang**: Division of Hematology and Oncology, Department of Medicine, University of California San Francisco, San Francisco, United States; **Kerwyn Casey Huang**: Department of Bioengineering, Stanford University, Stanford, United States; **Juliana Idoyaga**: Department of Microbiology and Immunology, Stanford University, Stanford, United States; **Seung K Kim**: Department of Developmental Biology, Stanford University School of Medicine, Stanford, United States; **Mark Krasnow**: Howard Hughes Medical Institute, Chevy Chase, United States; **Christin Kuo**: Department of Pediatrics - Pulmonary Medicine, Stanford University, Stanford, United States; **Patricia Nguyen**: Department of Medicine, Division of Cardiovascular Medicine, Stanford University, Stanford, United States; Veterans Administration Palo Alto Health Care System and Department of Medicine, Palo Alto, United States; Stanford Cardiovascular Institute, Stanford, United States; **Thomas A Rando**: Paul F. Glenn Center for the Biology of Aging, Stanford University School of Medicine, Stanford, United States; **Kristy Red-Horse**: Department of Biology, Stanford University, Stanford, United States; **Jeremy Reiter**: Department of Biochemistry, University of California San Francisco, San Francisco, United States; **Justin Sonnenburg**: Department of Microbiology and Immunology, Stanford University, Stanford, United States; **Albert Wu**: Department of Ophthalmology, Stanford University School of Medicine, Stanford, United States; **Sean Wu**: Department of Medicine, Division of Cardiovascular Medicine, Stanford University, Stanford, United States; Stanford Cardiovascular Institute, Stanford, United States; **Tony Wyss-Coray**: Paul F. Glenn Center for the Biology of Aging, Stanford University School of Medicine, Stanford, United States

## Funding

| Funder | Grant reference number | Author |
| --- | --- | --- |
| National Science Foundation | DGE-1656518 | Julia Eve Olivieri |

| Funder | Grant reference number | Author |
|---|---|---|
| National Institute of General Medical Sciences | R01 GM116847 | Julia Salzman |
| National Science Foundation | MCB1552196 | Julia Salzman |
| National Institutes of Health | T15 LM7033-36 | Roozbeh Dehghannasiri |
| National Cancer Institute | R25 CA180993 | Roozbeh Dehghannasiri |

The funders had no role in study design, data collection and interpretation, or the decision to submit the work for publication.

### Author contributions

Julia Eve Olivieri, Roozbeh Dehghannasiri, Conceptualization, Data curation, Formal analysis, Investigation, Methodology, Resources, Software, Validation, Visualization, Writing – original draft; Peter L Wang, Investigation, Methodology, Validation, Visualization; SoRi Jang, Antoine de Morree, Serena Y Tan, Formal analysis, Investigation, Validation; Jingsi Ming, Angela Ruohao Wu, Data curation, Resources; Tabula Sapiens Consortium, Data curation; Stephen R Quake, Mark A Krasnow, Project administration, Resources, Supervision; Julia Salzman, Conceptualization, Formal analysis, Funding acquisition, Investigation, Methodology, Project administration, Resources, Supervision, Writing – original draft, Writing – review and editing

### Author ORCIDs
Julia Eve Olivieri (ID) http://orcid.org/0000-0002-0850-5498
Roozbeh Dehghannasiri (ID) http://orcid.org/0000-0001-7413-3437
Peter L Wang (ID) http://orcid.org/0000-0001-9651-3860
Antoine de Morree (ID) http://orcid.org/0000-0002-8316-4531
Julia Salzman (ID) http://orcid.org/0000-0001-7630-3436

### Decision letter and Author response
Decision letter https://doi.org/10.7554/eLife.70692.sa1
Author response https://doi.org/10.7554/eLife.70692.sa2

## Additional files

### Supplementary files
• Supplementary file 1. Dataset summary. Brief overview of the datasets used in this paper, including tissues analyzed, number of cells, median number of spliced reads per cell, sex, and age.

• Supplementary file 2. Differential alternative splicing per compartment. Separate table with the p-value based on the SpliZ and SpliZVD for each gene in each dataset, testing differences between compartments. The gene name, SpliZ p-value, SpliZVD p-value, and largest magnitude median for all compartments for SpliZ and SpliZVD for each gene are given by the `geneR1A_uniq`, `perm_pval_adj_scZ`, `perm_pval_adj_svd_z0`, `max_abs_median_scZ`, and `max_abs_median_svd_z0` columns, respectively. (A) Human individual 1 10X ; (B) human individual 2 10 X; (C) human individual 1 SS2; (D) human individual 2 SS2; (E) lemur individual 1 10 X; (F) lemur individual 2 10X ; (G) mouse individual 1 10X ; (H) mouse individual 2 10X .

• Supplementary file 3. Differential alternative splicing per cell type. Separate table with the p-values based on the SpliZ and SpliZVD for each gene in each dataset, testing differences between cell types. The gene name, SpliZ p-value, SpliZVD p-value, and largest magnitude median for all compartments for SpliZ and SpliZVD for that gene are given by the geneR1A_uniq, perm_pval_adj_scZ, perm_pval_adj_svd_z0, max_abs_median_scZ, and max_abs_median_svd_z0 columns, respectively. (A) human individual 1 10X ; (B) human individual 2 10X; (C) human individual 1 SS2; (D) human individual 2 SS2; (E) lemur individual 1 10 X; (F) lemur individual 2 10X ; (G) mouse individual 1 10X ; (H) mouse individual 2 10X .

• Supplementary file 4. Most variable splice sites (SpliZsites). The most variable splice sites (SpliZsites) for genes with significant alternative splicing in human individual 1 10X  data. Each line reports a SpliZsite and contains the coordinates, whether it is an annotated exon, whether it is an

exon with known alternative splicing, whether the splice site is in 5′ or 3′ UTR of the gene, and whether the SpliZsite found to be a SpliZsite in mouse lemur and mouse datasets.

• Supplementary file 5. Regulated alternative splicing events in spermatogenesis. The list of genes with significantly regulated alternative splicing during sperm development. Each line contains information about the number of cells, Spearman's correlation, and its p-value for the human gene and also the same information (based on the mouse and mouse lemur sperm data) for its orthologous genes in mouse and mouse lemur genomes.

• Transparent reporting form

### Data availability

The fastq files for the Tabula Sapiens data (both 10X Chromium and Smart-seq2) were downloaded from https://tabula-sapiens-portal.ds.czbiohub.org/. The pilot 2 individual is referred to as individual 1, and the pilot 1 individual is referred to as individual 2 in this manuscript. Pancreas data was removed from individual 2. Cell type annotations were downloaded on March 19th, 2021, and the "ground truth" column was used as the within-tissue-compartment cell type. The Tabula Muris data was downloaded from a public AWS S3 bucket according to https://registry.opendata.aws/tabula-muris-senis/. The P1 (30-M-2) mouse is referred to as individual 1 and P2 (30-M-4) is referred to as individual 2 in this manuscript. Compartment annotations were assigned based on knowledge of cell type. The fastq files for the Tabula Microcebus mouse lemur data were downloaded from https://tabula-microcebus.ds.czbiohub.org. Mouse lemurs 4 and 2 are referred to as individuals 1 and 2, respectively, in this manuscript. The propagated_cell_ontology_class column was used as the within-tissue-compartment cell type. Because tissue compartments in the mouse lemur were annotated more finely, we collapsed the lymphoid, myeloid, and megakaryocyte-erythroid compartments into the immune compartment. Human and mouse unselected spermatogenesis data was downloaded from the SRA databases with accession IDs SRR6459190 (AdultHuman_17-3), SRR6459191 (AdultHuman_17-4), and SRR6459192 (AdultHuman_17-5) for human, and accession IDs SRR6459155 (AdultMouse-Rep1), SRR6459156 (AdultMouse-Rep2), and SRR6459157 (AdultMouse-Rep3) for mouse. The files containing SpliZ values can be accessed at the following FigShare repository: https://doi.org/10.6084/m9.figshare.14531721.

The following dataset was generated:

| Author(s) | Year | Dataset title | Dataset URL | Database and Identifier |
|---|---|---|---|---|
| Olivieri JO | 2021 | RNA splicing programs define tissue compartments and cell types at single cell resolution | https://figshare.com/articles/dataset/RNA_splicing_programs_define_tissue_compartments_and_cell_types_at_single_cell_resolution/14531721 | figshare, 10.6084/m9.figshare.14531721 |

The following previously published datasets were used:

| Author(s) | Year | Dataset title | Dataset URL | Database and Identifier |
|---|---|---|---|---|
| Tabula Microcebus Consortium | 2021 | Tabula Microcebus | https://tabula-microcebus.ds.czbiohub.org/about | Tabula Microcebus, czbiohub |
| Tabula Muris Consortium | 2018 | Tabula Muris | https://tabula-muris.ds.czbiohub.org/ | Tabula Muris, ds.czbiohub |
| Tabula Sapiens Consortium | 2021 | Tabula Sapiens | https://tabula-sapiens-portal.ds.czbiohub.org/ | Tabula Sapiens, portal.ds.czbiohub |

*Continued on next page*

*Continued*

| Author(s) | Year | Dataset title | Dataset URL | Database and Identifier |
|---|---|---|---|---|
| Hermann BP | 2018 | AdultHuman_17-3 | https://www.ncbi.nlm.nih.gov/geo/query/acc.cgi?acc=SRR6459190 | NCBI Gene Expression Omnibus, SRR6459190 |
| Hermann BP | 2018 | AdultHuman_17-4 | https://www.ncbi.nlm.nih.gov/geo/query/acc.cgi?acc=SRR6459191 | NCBI Gene Expression Omnibus, SRR6459191 |
| Hermann BP | 2018 | AdultHuman_17-5 | https://www.ncbi.nlm.nih.gov/geo/query/acc.cgi?acc=SRR6459192 | NCBI Gene Expression Omnibus, SRR6459192 |
| Hermann BP | 2018 | AdultMouse-Rep1 | https://www.ncbi.nlm.nih.gov/geo/query/acc.cgi?acc=SRR6459155 | NCBI Gene Expression Omnibus, SRR6459155 |
| Hermann BP | 2018 | AdultMouse-Rep2 | https://www.ncbi.nlm.nih.gov/geo/query/acc.cgi?acc=SRR6459156 | NCBI Gene Expression Omnibus, SRR6459156 |
| Hermann BP | 2018 | AdultMouse-Rep3 | https://www.ncbi.nlm.nih.gov/geo/query/acc.cgi?acc=SRR6459157 | NCBI Gene Expression Omnibus, SRR6459157 |

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
