## [Decision Letter]

**Acceptance summary:**

This study describes an analysis of cell type-specific alternative splicing using 10x scRNA-seq data. This work shows that in spite of the challenges associated with the analysis of such datasets, it is possible to identify alternative exons with differential splicing between tissue compartments and to some extent reveal cell types by splicing profiles of single cells. This work is informative regarding what can be done to analyze alternative splicing using 10X data and fills in a gap in the field. Your revised manuscript addresses reviewers' concerns and strengthen the manuscript for the general audience and we are most appreciative of it.

**Decision letter after peer review:**

Thank you for submitting your article "RNA splicing programs define tissue compartments and cell types at single cell resolution" for consideration by *eLife*. Your article has been reviewed by 2 peer reviewers, one of whom is a member of our Board of Reviewing Editors, and the evaluation has been overseen by Patricia Wittkopp as the Senior Editor. The reviewers have opted to remain anonymous.

Essential revisions:

1) The computational analysis appears to be solid in general but the presentation, as in the current form, need to be improved before publication.

2) Overall, there is little doubt that alternative exons near the 3' end of the transcripts can be studied by 10X data, but the scope is relatively limited. This is confirmed in this study, as only 1353 genes can be quantified at the exon level and only 22 genes were identified to have differential splicing. The study needs to be very clearly discuss this major limitation and balance the pros and cons of their method.

3) The algorithm is under review at another journal and has made the review process here difficult (several reviewers bowed out for this reason) – thus is is very important for SpliZ to be thoroughly discussed here. It would actually be even better if the paper describing the method were accepted for publication prior to publication of this work.

4) Some claims are made that are not fully substantiated by the data, which could be addressed by adding more context to the analyses. Furthermore, some figures can be tweaked or expanded upon in the text to improve clarity. In particular, the following sections should be amended:

"Supporting the idea that SpliZsites 31 discover real biological signal, 13% of the SpliZsites in the human are also identified as SpliZsites in the mouse lemur and/or mouse (Methods)."

Insufficient context is provided here for readers for this conclusion. The authors should compare this number to a shuffled dataset or other background data to demonstrate the supposed significance of the 13% number provided.

*Reviewer #1 (Recommendations for the authors):*

Some claims are made that are not fully substantiated by the data, which could be addressed by adding more context to the analyses. Furthermore, some figures can be tweaked or expanded upon in the text to improve clarity. In particular, the following sections should be amended:

"Supporting the idea that SpliZsites 31 discover real biological signal, 13% of the SpliZsites in the human are also identified as SpliZsites in the mouse lemur and/or mouse (Methods)."

Insufficient context is provided here for readers for this conclusion. The authors should compare this number to a shuffled dataset or other background data to demonstrate the supposed significance of the 13% number provided.

In Figure 2B the authors demonstrate prediction of cellular compartment of cells using k-means clustering analysis on SpliZ scores from two genes. The claim in the main text: "Setting k=3, cells from stromal, epithelial, and immune compartments were classified with accuracies of 78%, 84%, and 95% respectively independent of gene expression". Though these were the results from one of the two individuals in the dataset, the accuracies were much worse for the other individual, and the text is misleading here in only focusing on the cleaner data. The authors should acknowledge this in the text, and address the possible factors causing lower accuracy in individual 2. Furthermore, given the cells come from 4 tissue compartments (immune, epithelial, endothelial and stromal), the authors should elaborate on the decision to set k to 3.

Figure 3A, a legend for the squares and the circles, indicating 10X and Smart-seq can be more clear. (like those in Supp Figure 2)

Figure 7B, though the splicing changes are correlated across the species, the directionality of splice site usage is inverted across species. The authors can discuss more on the biological meaning.

*Reviewer #2 (Recommendations for the authors):*

The computational analysis appears to be solid in general but the presentation, as in the current form, need to be improved before publication.

1. Overall, I do not have doubt that alternative exons near the 3' end of the transcripts can be studied by 10X data, but the extend is relatively limited. This is confirmed in this study, as only 1353 genes can be quantified at the exon level and only 22 genes were identified to have differential splicing. I think despite the limitation, efforts to mine splicing using 10X data should still be encouraged, given the explosive number of datasets available. However, the discussion of the pro and cons should be balanced (e.g., the 3' bias should be discussed).

2. The authors used a new pipeline named SpliZ for analysis (preprint cited as ref. 26), and several variations SpliZsite and SpliZVD were also used. While it is fine to present technical details in separate publications, key features have to be described in the manuscript, which is required for understanding the results. For example, what does SpliZ measure (something similar to PSI I assume), what is SpliZsite (filtered splice sites from STAR alignments?), what is the difference beteen SpliZ and SpliZVD? How dropout and UMIs are handled in the pipeline?

3. Insufficient descriptions were provided in multiple figures.

Figure 2A, circles and squares were not explained (they were explained in Figure 3 below). How the dot plots are related to the splice sites shown in the sashimi plots and how the splice sites in the sashimi plots are related to the gene structure schematics (need some guesswork)? What is shown in the boxplot (labelled "Average 3' splice site per cell", which I assume it SpliZ score, but again is this the fraction of reads that uses the upstream 3' splice site?)

Figure 2C, D. Is each dot a single cell or a "meta cell" that averages a certain number of individual cells?

Figure 3A, how the gene structure schematic relate to the boxplot above is confusing. Also in the gene schematics of the three species further down, it is confusing why two parts of the human gene were highlighted (only the two 3' splice sites near the 3' end are relevant?). Unclear what is shown in the illustrations on the right of the gene schematics.

Figure 3D. How exons labeled 5,6,7 are related to site 1 and 2 in A-C (need guesswork)?

Figure 4. Only read fraction is shown but not SpliZ scores?

Figure 5A, similar to the question for Figure 2A, how the gene structure schematic relate to the boxplot above is confusing.

Figure 6C. It does not seem to be the evidence of a single exon to distinguish two subpopulations of monocytes are convincing. How do we know whether the two subpopulations reflect certain technical issues (potentially similar to the controversial "bimodal splicing" proposed in previous publications)?

Similar confusion in Figure 7 as in Figures 2 and 5.

4. I thought some global evaluation on the reliability of 10X results using independent datasets will be important (e.g., correlation of differential splicing in comparison with results from bulk RNA-seq and/or SMART-seq data). Some of the results presented in Supplementary Figures(e.g., Figure S6) can probably presented as the main figure.

---

## [Author Response]

Essential revisions:1) The computational analysis appears to be solid in general but the presentation, as in the current form, need to be improved before publication.

We are pleased to hear that you find the analysis solid in general. We have now addressed all suggestions and comments made by the reviewers to greatly improve the presentation of this manuscript for publication, including revising figures to make interpretation more clear.

2) Overall, there is little doubt that alternative exons near the 3' end of the transcripts can be studied by 10X data, but the scope is relatively limited. This is confirmed in this study, as only 1353 genes can be quantified at the exon level and only 22 genes were identified to have differential splicing. The study needs to be very clearly discuss this major limitation and balance the pros and cons of their method.

We appreciate the concern that the limited scope of 10X data for splicing discovery was not emphasized enough in the original draft. To clarify, 22 genes were found to have differential compartment-specific splicing, while 129 genes were found to have cell-type-specific differential splicing out of 1,416 genes (page 6). We have added further clarification of the pros and cons of our method on page 9 in the revised manuscript:

“Although the SpliZ method enables biological discovery of splicing differences based on droplet-based sequencing data, droplet-based data still presents major challenges for splicing analysis compared to full-length data. In this study, droplet-based sequencing has much lower sequencing coverage than full-length data, resulting in only 1,416 genes with measurable SpliZ values in the first human individual based on 10X data compared to 9,802 genes with measurable SpliZ values in SS2 data. Additionally, current droplet-based data is 3-prime-biased, meaning that some splicing events will never be sequenced by the technology and therefore cannot be analyzed. Despite these challenges, the ubiquity of droplet-based data, its utility for profiling rare cell types, and its unprecedented scale make it a useful resource for splicing analysis.”

3) The algorithm is under review at another journal and has made the review process here difficult (several reviewers bowed out for this reason) – thus is is very important for SpliZ to be thoroughly discussed here. It would actually be even better if the paper describing the method were accepted for publication prior to publication of this work.

We understand that the SpliZ paper not being published makes it more difficult to review this manuscript. The methods paper for the SpliZ is currently in review at Nature Methods. It has been recently reviewed and based on the positive reviewers’ comments the Editor invited us for resubmission. We have now submitted the revision and it is currently being re-evaluated by reviewers. We are happy to share the comments from the reviewers on that manuscript if it would help you make a decision. We have also now added a thorough explanation of the SpliZ to the methods in a new section called “Explanation of the SpliZ method” on page 12 and added several more sentences of explanation to the main text at top of page 3:

“A large negative (resp. positive) SpliZ score for a gene in a cell means that the cell has shorter (resp. longer) introns than average for that gene. In the simplest exon skipping case, the SpliZ reduces to PSI.”

4) Some claims are made that are not fully substantiated by the data, which could be addressed by adding more context to the analyses. Furthermore, some figures can be tweaked or expanded upon in the text to improve clarity.

We are grateful for the thoughtful reading of the paper that revealed the lack of clarity in some analyses and figures. We have now added more context everywhere it was requested.

In particular, the following sections should be amended:"Supporting the idea that SpliZsites 31 discover real biological signal, 13% of the SpliZsites in the human are also identified as SpliZsites in the mouse lemur and/or mouse (Methods)."Insufficient context is provided here for readers for this conclusion. The authors should compare this number to a shuffled dataset or other background data to demonstrate the supposed significance of the 13% number provided.

We appreciate the point that the 13% fraction is provided without enough context for the reader to understand its significance. To add more clarity to our analysis, we now report the fraction of shared SpliZsites with mouse lemur and mouse separately. We have now changed the text on page 7 as follows:

“15.5% of LiftOver human SpliZsites were also SpliZsites in the mouse lemur compared to 7% expected under the null (Methods). Only 8.0% of LiftOver SpliZsites in human were called as SpliZsites in mouse compared to 8.8% expected under the null. This could be due to many factors including a larger evolutionary distance between mouse and human, smaller number of analyzed mouse cells, or lower sequencing depth (Suppl. Table 1)”

We also provided the details of our statistical analysis in a new section “SpliZsite analysis” in the Methods on page 16. We should point out that SpliZsites between organisms are not completely comparable due to lack of perfect LiftOver mappings between organisms and also different read depth and number of cells for the same gene in different organisms which can lead to a SpliZsite for a gene not being detected in all organisms.

Reviewer #1 (Recommendations for the authors):Some claims are made that are not fully substantiated by the data, which could be addressed by adding more context to the analyses. Furthermore, some figures can be tweaked or expanded upon in the text to improve clarity. In particular, the following sections should be amended:"Supporting the idea that SpliZsites 31 discover real biological signal, 13% of the SpliZsites in the human are also identified as SpliZsites in the mouse lemur and/or mouse (Methods)."Insufficient context is provided here for readers for this conclusion. The authors should compare this number to a shuffled dataset or other background data to demonstrate the supposed significance of the 13% number provided.

We appreciate the point that the 13% fraction is provided without enough context for the reader to understand its significance. To add more clarity to our analysis, we now report the fraction of shared SpliZsites with mouse lemur and mouse separately. We have now changed the text on page 7 as follows:

“15.5% of LiftOver human SpliZsites were also SpliZsites in the mouse lemur compared to 7% expected under the null (Methods). Only 8.0% of LiftOver SpliZsites in human were called as SpliZsites in mouse compared to 8.8% expected under the null. This could be due to many factors including a larger evolutionary distance between mouse and human, smaller number of analyzed mouse cells, or lower sequencing depth (Suppl. Table 1)”

We also provided the details of our statistical analysis in a new section “SpliZsite analysis” in the Methods on page 16. We should point out that SpliZsites between organisms are not completely comparable due to lack of perfect LiftOver mappings between organisms and also different read depth and number of cells for the same gene in different organisms which can lead to a SpliZsite for a gene not being detected in all organisms.

In Figure 2B the authors demonstrate prediction of cellular compartment of cells using k-means clustering analysis on SpliZ scores from two genes. The claim in the main text: "Setting k=3, cells from stromal, epithelial, and immune compartments were classified with accuracies of 78%, 84%, and 95% respectively independent of gene expression". Though these were the results from one of the two individuals in the dataset, the accuracies were much worse for the other individual, and the text is misleading here in only focusing on the cleaner data. The authors should acknowledge this in the text, and address the possible factors causing lower accuracy in individual 2. Furthermore, given the cells come from 4 tissue compartments (immune, epithelial, endothelial and stromal), the authors should elaborate on the decision to set k to 3.

Thank you for pointing this out. We have now added the accuracies for the second individual to the main text, included a discussion of why the accuracies may be worse in the second individual, and moved the explanation of not including the endothelial compartment from the methods to the main text (see paragraph 2 on page 4):

“Setting k=3, cells from stromal, epithelial, and immune compartments were classified with accuracies of 78%, 84%, and 95% respectively independent of gene expression in the first human individual (70%, 100%, and 49% in the second individual) (Figure 2B-D, Methods). The lower accuracy for individual 2 may be caused by individual 2 having only a third as many cells. The endothelial compartment was not included because it had a small proportion of cells in both datasets (3.7% in individual 1, 4.5% in individual 2).”

Figure 3A, a legend for the squares and the circles, indicating 10X and Smart-seq can be more clear. (like those in Supp Figure 2)

Thank you for your comment. We have now added a legend for Figure 3A (to the right of panel C) in the revised figure.

Figure 7B, though the splicing changes are correlated across the species, the directionality of splice site usage is inverted across species. The authors can discuss more on the biological meaning.

Thank you for pointing this out. The reason for the opposite direction of the splicing changes is that in human, *CEP112* is one the minus strand but it is on the plus strand in mouse and lemur genomes. We have now clarified this in the revised caption.

Reviewer #2 (Recommendations for the authors):The computational analysis appears to be solid in general but the presentation, as in the current form, need to be improved before publication.

Thank you for your detailed suggestions for how to improve the presentation of the paper. We have taken all feedback into account to make the figures and paper overall much clearer.

My major comments:1. Overall, I do not have doubt that alternative exons near the 3' end of the transcripts can be studied by 10X data, but the extend is relatively limited. This is confirmed in this study, as only 1353 genes can be quantified at the exon level and only 22 genes were identified to have differential splicing. I think despite the limitation, efforts to mine splicing using 10X data should still be encouraged, given the explosive number of datasets available. However, the discussion of the pro and cons should be balanced (e.g., the 3' bias should be discussed).

We appreciate the concern that the limited scope of 10X data for splicing discovery was not emphasized enough in the original draft. To clarify, 22 genes were found to have differential compartment-specific splicing, while 129 genes were found to have cell-type-specific differential splicing out of 1,416 genes (page 6). We have added further clarification of the pros and cons of our method to page 9 in the manuscript:

“Although the SpliZ method enables biological discovery of splicing differences based on droplet-based sequencing data, droplet-based data still presents major challenges for splicing analysis compared to full-length data. In this study, droplet-based sequencing has much lower sequencing coverage than full-length data, resulting in only 1,416 genes with measurable SpliZ values in the first human individual based on 10X data compared to 9,802 genes with measurable SpliZ values in SS2 data. Additionally, current droplet-based data is 3-prime-biased, meaning that some splicing events will never be sequenced by the technology and therefore cannot be analyzed. Despite these challenges, the ubiquity of droplet-based data, its utility for profiling rare cell types, and its unprecedented scale make it a powerful approach to discover regulated splicing.”

2. The authors used a new pipeline named SpliZ for analysis (preprint cited as ref. 26), and several variations SpliZsite and SpliZVD were also used. While it is fine to present technical details in separate publications, key features have to be described in the manuscript, which is required for understanding the results. For example, what does SpliZ measure (something similar to PSI I assume), what is SpliZsite (filtered splice sites from STAR alignments?), what is the difference beteen SpliZ and SpliZVD? How dropout and UMIs are handled in the pipeline?

We appreciate your comment on the lack of enough description for the SpliZ pipeline in the paper. We have now added a new section “Explanation of the SpliZ method” in the Methods to better describe the SpliZ pipeline on page 12. We have also added several more sentences of explanation to the main text at the beginning of page 3:

“A large negative (resp. positive) SpliZ score for a gene in a cell means that the cell has shorter (resp. longer) introns than average for that gene. In the simplest exon skipping case, the SpliZ reduces to PSI.”

In summary, we recommend SpliZ for 10x data (as it is less likely to capture multiple independent splicing events in a gene through 10x sequencing) and SpliZVD for the SmartSeq2 as it can systematically provide more strength for analyzing multiple splicing events by projecting them onto lower dimensions. We should also note that SplizSites are obtained as the splice sites that have the largest contribution to the overall variation in the genes found to be significantly regulated by the SpliZ and/or SpliZVD scores (we have now added a description for SPliZsites on page 3). SpliZ and SpliZVD are the only scores that could be used for finding genes with regulated splicing while SpliZsites is used to pinpoint the differential splicing pattern of a gene to one of its splice sites. Data is preprocessed with the SICILIAN pipeline (Dehghannasiri et al., 2021), which handles UMI deduplication (added on page 12).

3. Insufficient descriptions were provided in multiple figures.

We would like to thank you for pointing this out. We have now clarified the figures by adding more description to the legends, text, and to the figures themselves.

Figure 2A, circles and squares were not explained (they were explained in Figure 3 below).

Thank you for pointing out that circles and squares were not explained until figure 3. We have now moved that explanation up into the caption for figure 2.

How the dot plots are related to the splice sites shown in the sashimi plots and how the splice sites in the sashimi plots are related to the gene structure schematics (need some guesswork)? What is shown in the boxplot (labelled "Average 3' splice site per cell", which I assume it SpliZ score, but again is this the fraction of reads that uses the upstream 3' splice site?)

We appreciate your comment. The dot plots show the fraction of junctional reads for each splice site at the cell type level. The thickness of the sashimi arcs show the fraction of reads when all cells for the associated group of cell types (right below each sashimi) and all datasets (individuals and technologies) are considered together as pseudo bulk. Each vertical line for a splice site in the sashimi arc shows the fraction of reads corresponding to a 10x (circles) or SS2 (squares) dataset from a certain individual. The box plot shows the distribution of the average 3’ splice site for each cell within a cell type by assigning 1 and 2 to the closer and farther 3’ SS, respectively, and then computing their weighted average according to their number of junctional reads. We have now clarified this in the revised caption for Figure 2 and have also modified Figure 2A to make it clearer. We had also provided the SpliZ scores for *MYL6* in Figure 2-—figure supplement 1 (previously Suppl. Figure 1A). We have now mentioned this in the revised caption.

Figure 2C, D. Is each dot a single cell or a "meta cell" that averages a certain number of individual cells?

In Figures 2C, D, each dot shows the SpliZ score (Figure 2C) and the number of spliced reads (Figure 2D) for a single cell and dots are color coded according to the compartment of the cell. We have now clarified this in the revised caption for Figure 2.

Figure 3A, how the gene structure schematic relate to the boxplot above is confusing. Also in the gene schematics of the three species further down, it is confusing why two parts of the human gene were highlighted (only the two 3' splice sites near the 3' end are relevant?). Unclear what is shown in the illustrations on the right of the gene schematics.

We would like to thank you for your comment. To obtain the average 3’ splice site per cell, splice sites are ranked from 1 (closest to the 5’ SS) to 3 (farthest from the 5’ SS) and then the ranks were used to obtain the weighted average 3’ splice site per cell (based on the junctional reads from the 5’ splice site to each 3’ splice site). We put the schematic for protein domains in a separate panel (Figure 3E). The schematic shows how the 3 different protein domains in *MYL6* are organized in each *MYL6* isoform. We have now revised the caption to better describe the figure. We have also modified the figure itself to make it clearer.

Figure 3D. How exons labeled 5,6,7 are related to site 1 and 2 in A-C (need guesswork)?

Thank you for your comment. We have removed exon numbers in Figure 3D and instead have used consistent Refseq isoform IDs throughout different panels in Figure 3.

Figure 4. Only read fraction is shown but not SpliZ scores?

The SpliZ scores for genes *RPS24*, *MYL6*, and *ATP5F1C* are shown in Figure 2-—figure supplement 1-3 (previously Supplementary Figure 1). To visualize the scores across the entire cell population, we utilized the cellxgene software. In these figures, UMAP embedding is based on the gene expression but the cells in the second UMAP for each gene are color-coded according to the SpliZ score of the gene in each cell. We clarified this by adding the following text to the caption of Figure 3:

“The SpliZ scores (and also gene expression values) for *MYL6* across all 10x cells in human individual 1 are shown in Figure 2-—figure supplement 1”

Figure 5A, similar to the question for Figure 2A, how the gene structure schematic relate to the boxplot above is confusing.

We appreciate your comment. We have now clarified in the revised caption that we have used the same approach as in Figures 2 and 3 to obtain the box plots: ranking 3’ splice sites from 1 to 3 (from the closest one to the farthest one) and then obtaining the weighted average of 3’ splice site for each cell within a cell type. We have now modified Figure 6 (previously Figure 5) and added the following text to its caption to clarify this:

“The box plot shows the distribution of the average 3’ splice site (obtained as the weighted average of 3’ splice sites when ranked from 1 to 3 from the closest to the farthest according to their fraction of junctional reads) for the cells within a celltype (See Figures 2 and 3 for more explanation of dot and box plots.)”

Figure 6C. It does not seem to be the evidence of a single exon to distinguish two subpopulations of monocytes are convincing. How do we know whether the two subpopulations reflect certain technical issues (potentially similar to the controversial "bimodal splicing" proposed in previous publications)?

Thank you for bringing up the possibility that the distributions we’re seeing in the subpopulation analysis could be due to a technical artifact such as bimodal splicing (Buen Abad Najar et al., 2020). We agree that this is a big concern, and we appreciate the opportunity to provide more evidence for the reliability of the subclusters. We have now revised the first paragraph of page 8 and also added the following analysis into the Methods section on page 16 of the paper:

“To test whether the subpopulations of *SAT1* were the result of a “bimodal splicing” artifact as reported in (Buen Abad Najar, Yosef, and Lareau 2020), we performed the following analysis. Subsetting to only blood classical monocytes in human individual 1, we calculated the fraction of junctional reads *p* aligning to the 5’ splice site 23785328 in *SAT1* that partner with the 3’ splice site 23783883 rather than 23784403. We found *p* = 83/102 = 0.814 (the probability was 0.893 considering only individual 2). We then subset to only cells with exactly 2 reads mapping to 5’ splice site 23785328 in *SAT1*, resulting in 16 cells. Out of these 16 cells, all had either both reads mapping to 3’ splice site 23783883 or both reads mapping to 3’ splice site 23784403. We calculated the probability of zero cells having one read mapping to each splice site under the null hypothesis as follows: (1 – binom.pmf(1,2,0.814))^16^ = 0.00312 (binomial exact test). Because processing of 10X data includes a UMI deduplication step through SICILIAN (Dehghannasiri et al., 2021), these duplicates are not PCR duplicates.”

Similar confusion in Figure 7 as in Figures 2 and 5.

Thank you for pointing this out. The gray dashed lines between the gene structure and the dot plot show the corresponding splice site for each column of the dot plot and regarding box plots we similarly obtained the average 3’ splice site by ranking them from 1 to 4 and computing the weighted average of their ranks according to the number of junctional reads to each 3’ splice site. We have now better explained this in the revised caption for Figure 7 (now Figure 8).

4. I thought some global evaluation on the reliability of 10X results using independent datasets will be important (e.g., correlation of differential splicing in comparison with results from bulk RNA-seq and/or SMART-seq data). Some of the results presented in Supplementary Figures(e.g., Figure S6) can probably presented as the main figure.

We appreciate the interest in seeing global comparisons of the reliability of 10X results. We did not attempt a comparison of 10X with bulk data because bulk data represents a mixture of single cells with unknown proportions, which would confound the comparison between scRNA-Seq and bulk data. Instead, we validated two of our main discoveries, patterns in *MYL6* and *RPS24*, using FISH, a completely orthogonal method of validation. We have also already included global comparison with SS2 as you pointed out in Supplementary Figure 6, which we have now included as a main figure (Figure 5).